# Sum-of-Parts: Self-Attributing Neural Networks with End-to-End Learning of Feature Groups

**Weiqiu You** [1]  **Helen Qu** [2]  **Marco Gatti** [2]  **Bhuvnesh Jain** [2]  **Eric Wong** [1]

## Abstract

*Self-attributing neural networks (SANNs)* present a potential path towards interpretable models for high-dimensional problems, but often face significant trade-offs in performance. In this work, we formally prove a lower bound on errors of per-feature SANNs, whereas group-based SANNs can achieve zero error and thus high performance. Motivated by these insights, we propose *Sum-of-Parts (SOP)*, a framework that transforms any differentiable model into a group-based SANN, where feature groups are learned end-to-end without group supervision. SOP achieves state-of-the-art performance for SANNs on vision and language tasks, and we validate that the groups are interpretable on a range of quantitative and semantic metrics. We further validate the utility of SOP explanations in model debugging and cosmological scientific discovery. [1]

## 1  Introduction

Machine learning (ML) models are powerful at complex tasks, but also notoriously opaque. In high-stakes domains such as science (Li et al., 2021; Zednik & Boelsen, 2022) and medicine (Reyes et al., 2020; Tjoa & Guan, 2021), experts need explanations to trust the models' decisions. For instance, physicists use interpretable coefficients to validate an ML model's rediscovery of Kepler's first law (Li et al., 2021), while physicians require explanations to trust ML-driven diagnostic decisions (Klauschen et al., 2024).

---
[1]Department of Computer and Information Science, University of Pennsylvania, Philadelphia, PA, USA [2]Department of Physics and Astronomy, University of Pennsylvania, Philadelphia, PA, USA. Correspondence to: Weiqiu You <weiqiuy@seas.upenn.edu>, Eric Wong <exwong@seas.upenn.edu>.

*Proceedings of the $42^{nd}$ International Conference on Machine Learning*, Vancouver, Canada. PMLR 267, 2025. Copyright 2025 by the author(s).

[1]Our code is available at https://github.com/BrachioLab/sop

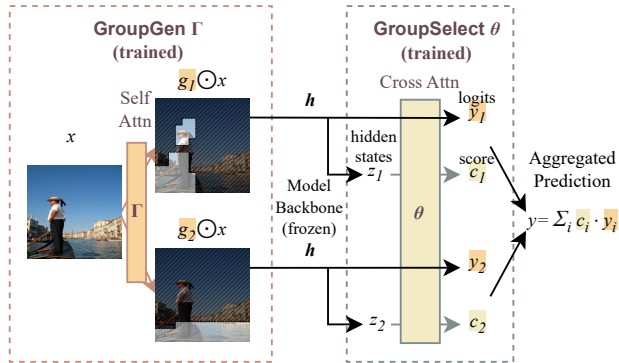

Figure 1. Sum-of-Parts (SOP) linearly aggregates outputs from multiple feature groups. This maintains performance while ensuring interpretability. SOP first generates groups using a *group generator* $\Gamma$, predicts with a pre-trained *backbone* $h$, and aggregates the group predictions with a *group selector* $\theta$.

Self-explaining neural networks (SENNs) were proposed as a way to create neural networks with guaranteed linear interpretations (Alvarez Melis & Jaakkola, 2018). Specifically, SENNs express predictions as linear combinations of interpretable atoms scaled by learnable coefficients, a natural generalization of the statistical interpretation of the classic linear model. These interpretable atoms represent semantic notions such as object segments in images or example image prototypes (Alvarez Melis & Jaakkola, 2018).

A common class of self-explaining neural networks, which we refer to as *Self-**Attributing** Neural Networks* (SANNs), use embedded feature subsets as interpretable atoms (Brendel & Bethge, 2019; Jain et al., 2020; Agarwal et al., 2021). SANNs have predictions faithfully decomposable as linear combinations of feature subset contributions. On the other hand, post-hoc feature attributions fail to pass the sanity checks for faithfulness (Sundararajan et al., 2017; Adebayo et al., 2018).

However, SANNs often exhibit performance trade-offs and rely on specific components such as per-feature modules in NAM (Agarwal et al., 2021), convolutional layers in BagNet (Brendel & Bethge, 2019), or attention mechanisms in FRESH (Jain et al., 2020). This architecture require-

ment hinders SANN from leveraging pre-trained models that achieve high performance on target tasks.

To understand these limitations, we first analyze a theoretical barrier that limits SANN's performance. Specifically, we prove a lower bound on the error of a class of self-attributing neural networks that we refer to as per-feature SANNs. This result shows that it is impossible for per-feature SANNs to achieve high performance when features are highly correlated, a critical limitation in high-dimensional vision and language data. In contrast, we further prove that group-based SANNs can achieve high performance in these settings, but require a careful selection of feature groups.

To overcome these provable limitations for per-feature SANNs and inflexible group-based SANNs, we propose **S**um-**of**-**P**arts (**SOP**), a flexible framework that transforms any differentiable model into a group-based SANN (Figure 1). Specifically, given a backbone model and an input, the framework (1) identifies feature groups with a learned attention module, (2) encodes each group using a model-agnostic backbone, and (3) aggregates predictions with a second sparse attention module. This framework can then be learned end-to-end with only the final prediction labels, notably without the direct supervision of feature groups. Learned feature groups can capture the dynamic correlations in high-dimensional data, enabling SOP to overcome the theoretical limitations of per-feature and fixed-group-based SANNs.

Our main contributions are as follows:

1. We propose Sum-of-Parts (SOP), a model-agnostic framework which transforms any model into a group-based SANN. The groups in SOP are end-to-end learnable without the need for group label supervision.

2. We formally prove that groups are essential for SANNs to achieve low errors for highly correlated features. In contrast, we prove a lower bound on per-feature SANNs' errors, which grows as the number of features increases.

3. We show that SOP achieves state-of-the-art performance among SANNs on vision and language tasks as informed by the theory, with learned interpretable groups validated on a range of quantitative and semantic metrics.

4. We validate the utility of SOP in debugging if correct/incorrect model predictions rely more on the background/objects, as well as a scientific discovery setting within cosmology by using the groups and scores to uncover new insights about galaxy formation.

## 2 Overcoming Self-Attributing Neural Networks' Limitations with Groups

In this section, we first review self-attributing neural networks (Section 2.1), and prove that the poor performance of previously explored per-feature SANNs is theoretically limited due to correlated features in high dimensional data (Section 2.2). In contrast, we further prove that group-based SANNs can overcome these fundamental limitations, motivating our proposed framework for learnable group-based SANNs (Section 2.3).

### 2.1 Self-Attributing Neural Networks

Self explaining neural networks model predictions as a linear combination $f(x) = \sum_i \theta(x)_i h(x)_i$, where $\theta(x)$ are linear coefficients and $h(x)$ are referred to as interpretable atoms (Alvarez Melis & Jaakkola, 2018). A common strategy for creating self-explaining neural networks is to use feature subsets as interpretable atoms. For example, Bag-Net (Brendel & Bethge, 2019) or Neural Additive Models (Agarwal et al., 2021) decompose predictions into a linear combination of terms, where each term is directly computed from and attributed to a subset of input features. We denote such models, which combine the interpretability of linearity with guaranteed attributions to input features, as Self-Attributing Neural Networks (SANNs).

**Definition 2.1.** A *self-attributing neural network* $f : \mathbb{R}^d \rightarrow \mathbb{R}$ given input $x \in \mathbb{R}^d$ decomposes predictions as follows:

$$f(x) = \sum_{i=1}^{m} \theta(x)_i h(x_{G_i}) \tag{1}$$

where $\theta(x) \in \mathbb{R}^m$ are linear coefficients, and $h(x_{G_i}) \in \mathbb{R}^m$ are embeddings of the feature subset $x_{G_i}$ corresponding to the subset $G_i \subseteq [d]$. Note that $m$ can be different from the number of raw features $d$.

The resulting linear combination constitutes a faithful-by-construction explanation for the decision process of the model (Lyu et al., 2024). SANNs are only as interpretable as the underlying feature subsets (Alvarez Melis & Jaakkola, 2018; Zytek et al., 2022), and different SANNs have explored various feature subsets $x_{G_i}$. For example, NAM (Agarwal et al., 2021) uses individual features, Bag-Net (Brendel & Bethge, 2019) relies on large patches, and FRESH (Jain et al., 2020) selects a single subset using attention scores. However, across these subsets, SANNs have consistently exhibit significant trade-offs in performance in exchange for interpretability. In this section, we theoretically analyze the underlying cause for this trade-off and how SANNs can overcome these barriers.

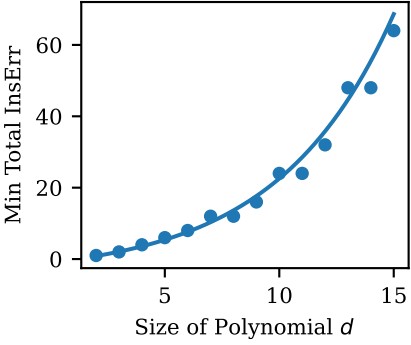

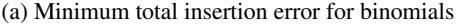

(a) Minimum total insertion error for binomials.

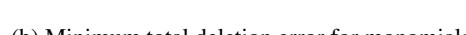

(b) Minimum total deletion error for monomials.

*Figure 2.* Errors for per-feature SANNs grow fast unavoidably. The minimum (a) total insertion error of monomials of size $d$ and (b) total deletion errors of binomials of size $d$ are the minima over all possible per-feature self-explaining models. The dots are the lower bounds computed by the solver, while the line is a best-fit exponential function.

## 2.2 Lower Bounds on the Error of Per-feature SANNs

One class of SANNs uses individual features as interpretable atoms (Agarwal et al., 2021), where each feature subset $G_i = \{i\}$ and the corresponding interpretable atom $h(x_{G_i}) = h(x_i)$ is the encoding of exactly one feature $x_i$. We begin by analyzing the error of per-feature SANNs.

In order for such models to be accurate, the contribution of a feature to the true label should match the change in prediction when the feature is removed or added. Similarly, the contribution of a subset, $\sum_{G_i \subseteq S} \alpha_i$, should capture the change in prediction when the subset $S$ is excluded or included. We formalize this difference between a SANN prediction and the ground truth when inserting or deleting subsets of features as insertion and deletion errors, respectively.

**Definition 2.2.** (Insertion Error) Let $\alpha_i = \theta(x)_i h(x_{G_i})$ be the total contribution of the $i$th feature group to the prediction of a SANN. Then, the *insertion error* of a self attributing neural network $f(x) = \sum_{i=1}^{m} \theta(x)_i h(x_{G_i}) = \sum_i \alpha_i$ for a target function $f^* : \mathbb{R}^d \to \mathbb{R}$ when inserting a subset of features $S$ to an input $x$ is

$$\mathrm{InsErr}(G, \alpha, S) = \left| f^*(x_S) - f^*(0_d) - \sum_{G_i \subseteq S} \alpha_i \right|$$

$$\text{where } (x_S)_j = \begin{cases} x_j & \text{if } j \in S \\ 0 & \text{otherwise} \end{cases}$$

The total insertion error over all possible insertions is $\sum_{S \subseteq [d]} \mathrm{InsErr}(G, \alpha, S)$.

The insertion error captures the difference between the ground truth effect of inserting a subset of features $S$ and the corresponding change in the SANN. If these two quantities are close, then the error is low. The insertion error of a per-feature SANN is a special case where $G = \{\{1\}, \ldots, \{d\}\}$.

For brevity of presentation, in this section we focus on insertion, and present analogous definitions and theorems for deletion to Appendix A. We note that the insertion and deletion procedures are analogous to insertion and deletion tests for post-hoc explanations (Petsiuk et al., 2018; Samek et al., 2017), but used here to capture the error of a SANN.

**Error Lower Bounds for Data with Correlated Features.** We now prove that it is impossible for per-feature SANNs to perform well when the data contains correlated features. Specifically, in Theorem 2.3, we show that when estimating polynomials function with correlated features, per-feature SANNs have a non-trivial lower bound on the total insertion error.

**Theorem 2.3** (Lower Bound on Insertion Error for Binomials). *Let $p : \{0,1\}^d \to \{0,1,2\}$ be a multilinear binomial polynomial function. Furthermore suppose that the features can be partitioned into $(S_1, S_2, S_3)$ of equal sizes where $p(x) = \prod_{i \in S_1 \cup S_2} x_i + \prod_{j \in S_2 \cup S_3} x_j$. Then, $\sum_{S \subseteq [d]} \mathrm{InsErr}(G, \alpha, S) \geq D_{ins}(\hat{\lambda})$, where $D_{ins}(\hat{\lambda}) = (\hat{\lambda}_1 - \hat{\lambda}_2)^\top c$ is the lower bound, $\hat{\lambda}$ is a dual feasible point, and $c$ is a constant as defined in (16).*

To derive this lower bound, we formulated the minimum total error of any SANN as a linear program, and used a dual feasible point to compute a lower bound. The proof and an analogous theorem for deletion error on monomials (Theorem A.2) are presented in Appendix A.

**Lower Bounds Grow Rapidly with Dimension.** In Figure 2, we calculate the lower bound using ECOS (Domahidi et al., 2013) and plot the lower bound for total insertion and deletion errors as the feature dimension grows (Figure 2), and observe that these errors increase exponentially with $d$. Altogether, these theoretical lower bounds and empirical trends suggest that per-feature SANNs are fundamentally in-

---

**Algorithm 1** The Sum-of-Parts Framework

---

**Require:** Backbone Pred $h$, Backbone Encoder $h_h$, Embedding Model $h_e$, Class Weights $C_h$
**Require:** GroupGen $\Gamma$, GroupSelect $\theta$, Number of Groups $m$, Input $x$

$(g_1, \ldots, g_m) \leftarrow \Gamma(x) \leftarrow \text{SoftSelfAttn}_{\tau=0.2}(h_e(x))$      {Group Generating (**trained**)}

  **for** $i = 1 \rightarrow m$ **do**
    $y_i \leftarrow h(g_i \odot x)$      {Predicting with *frozen* Backbone}
  **end for**

$z \leftarrow (h_h(g_1 \odot x), \ldots, h_h(g_m \odot x))$

$(c_1, \ldots, c_m) \leftarrow \theta(\Gamma(x), x) \leftarrow \text{SparseCrossAttn}(C_h, z)$      {GroupSelect (**trained**)}

$y \leftarrow \sum_{i=1}^{m} c_i y_i$

---

capable of modeling high-dimensional data, as they cannot model even simple polynomials.

### 2.3 Group-based Self-Attributing Neural Networks Can Overcome the Performance Barrier

The fundamental limitation of per-feature SANNs comes from its choice of interpretable atom: individual features are unable to capture correlations between multiple features. If we allow SANNs to use more expressive interpretable atoms composed of feature groups, can we get past this limitation? To answer this question, we carry out an analogous analysis for more general, group-based SANNs.

In this section, we summarize our main theoretical result in Theorem 2.4: we prove that there exist group-based SANNs that can not only perfectly capture the earlier settings in Theorem 2.3 (and Theorem A.2 in the appendix), but also far more complex, general polynomials with *zero* error. In other words, the right groups can enable SANNs to capture correlated signals and overcome the performance barrier.

**Theorem 2.4** (Informal: Zero Group Insertion and Deletion Error)**.** *For any general $m$-nomial polynomial $p$, having at most $m$ groups is sufficient for a group-based self-attributing neural network to achieve zero insertion and deletion error. See Theorem A.5 for full theorem and proof.*

Intuitively, a SANN can achieve low error if its groups align with the correlations in the data. Specifically, consider data generated from a polynomial with multiple terms $p(x) = q_1(x_{G'_1}) + \ldots q_m(x_{G'_m})$, where each $q_i(x_{G'_i})$ is a multiplicative term that depends on the group of features in $G'_i$. Then, a group-based SANN can achieve low error if each correlated feature group $G'_i$ aligns with a SANN group $G_i$. On the other hand, misaligned or insufficiently many groups lead to nonzero errors, highlighting how group-alignment is key to SANN performance.

While this theorem demonstrates that SANNs can model highly complex polynomials, it also provides insight into why existing SANNs have suffered major performance trade-

offs. For example, SANNs that rely on rigid patterns (Brendel & Bethge, 2019), use groups that are too small (Agarwal et al., 2021) or use too few groups (Jain et al., 2020) are unlikely to align with the ground truth correlations. In contrast, a high-performing SANN requires the ability to use many groups of flexible patterns to capture the diverse signals in high-dimensional data. These criteria motivate a new type of SANN that can overcome this theoretical performance barrier: the Sum-of-Parts framework.

## 3 The Sum-of-Parts Framework

In this section, we introduce our main technical contribution Sum-of-Parts (SOP), a framework that transforms any differentiable model into a group-based self-attributing model.

Suppose we have an input $x \in \mathbb{R}^d$ and a backbone model $h : \mathbb{R}^d \rightarrow \mathbb{R}$ that makes a prediction with the input, and we hope to convert $h$ to a SANN. A SANN requires components to generate and encode feature subsets and another component to assign them coefficients.

SOP therefore naturally consists of three parts: a **group generator** $\Gamma : \mathbb{R}^d \rightarrow \{0,1\}^{m \times d}$ that generates groups $g_1, \ldots, g_m \in \{0,1\}^d$, a **backbone predictor** $h : \mathbb{R}^d \rightarrow \mathbb{R}$ that makes a prediction with each group of features, and a **group selector** $\theta : \{0,1\}^{m \times d} \times \mathbb{R}^d \rightarrow [0,1]^m$ that assigns scores to the groups:

$$f(x) = \sum_{i=1}^{m} \underbrace{\theta(\Gamma(x), x)_i}_{\substack{\text{group} \\ \text{selector} \\ \textbf{(trained)}}} \cdot \underbrace{h(g_i \odot x)}_{\substack{\text{backbone} \\ \text{predictor} \\ \textit{(frozen)}}}, \quad \text{where } \underbrace{g_i \in \Gamma(x)}_{\substack{\text{group} \\ \text{generator} \\ \textbf{(trained)}}}.$$

Here we consider a single predicted logit, while the process can be repeated in batch for multiple logits or classes. Our algorithm is illustrated in Figure 1 and Algorithm 1.

**Group Generator** $\Gamma : \mathbb{R}^d \rightarrow \{0,1\}^{m \times d}$ takes in an input $x \in \mathbb{R}^d$ and outputs $m$ binary group masks $g_1, \ldots, g_m \in$

$\{0, 1\}^d$, where $g_{ij} = 1$ if and only if the feature $j$ is included in group $g_i$.[2] It uses a multi-headed self-attention module (Vaswani et al., 2017) to assign scores to features, and threshold each attention distribution to include top $\tau = 20\%$ features to each group.

$$\Gamma(x) = (g_1, \ldots, g_m) = \text{SoftSelfAttn}_{\tau=20\%}(h_e(x))$$

where the encoder $h_e : \mathbb{R}^d \to \mathbb{R}^{d \times e}$, which we typically take up to the penultimate layer of the backbone model, embeds each feature $x_i$ into a vector with embedding dimension $e$. The *learnable* group generator dynamically creates feature groups for each input, enabling better correlation compared to fixed groups (e.g., patches). Moreover, it has no specific architectural constraints on the backbone such as attention mechanisms.

**Backbone Predictor** $h : \mathbb{R}^d \to \mathbb{R}$ then makes a prediction with the input $x$ masked by each binary group mask $g_i$:

$$y_i = h(g_i \odot x), \quad i = 1, \ldots, m$$

where $y_i \in \mathbb{R}$ is the output logit and $\odot$ is Hadamard product. The backbone predictor can be *arbitrary* high-performing pre-trained model.

**Group Selector** $\theta : \{0, 1\}^{m \times d} \times \mathbb{R}^d \to [0, 1]^m$ then takes in the encoding of each group and uses a sparse cross-attention module to assign each group a score.

$$\theta(\Gamma(x), x) = (c_1, \ldots, c_m) = \text{SparseCrossAttn}(C_h, z)$$

where the query $C_h \in \mathbb{R}^k$ is initialized using the target class's weights with $k$ hidden dimensions and the key $z = (h_h(g_1 \odot x), \ldots, h_h(g_m \odot x)) \in \mathbb{R}^{m \times k}$ are last hidden states of all groups. As using a sparse number of groups avoids overloading human users, we replace the softmax in the cross attention with a sparse variant, the sparsemax operator (Martins & Astudillo, 2016). *Dynamically* assigning scores allows the model to choose the most helpful groups for prediction, while the *sparse* number of groups ensures easy human interpretability.

**The final prediction** is made by aggregating predictions from each group $g_i$ with its assigned score $c_i$.

$$f(x) = y = c_1 y_1 + \cdots + c_m y_m$$

To address the gradient flow issue caused by binarized groups, we incorporate a scaling factor based on the attention score in the final loss, as detailed in Appendix B.3. Additional details on self-attention, cross-attention, and embedding models are provided in Appendix B.

In summary, the learnable group generator dynamically creates correlated groups needed for high-performing SANNs, the model-agnostic framework supports arbitrary backbone predictors, and the sparse group selector assigns contributions for a small number of groups. Together, these components are essential for SOP to be a high-performing SANN, as informed by theory in Section 2.

# 4 Empirical Evaluations of Sum-of-Parts

We conduct experiments using our framework on image and text tasks to see if our theory-informed framework actually uses the learned groups for **(RQ1)** high *performance*. Next, we quantitatively measure the interpretability of SOP with **(RQ2)** the performance at different inference-time *sparsity* levels, **(RQ3)** various *faithfulness* metrics, and **(RQ4)** whether the group masks are *leaking* predictive signals.

Then, we measure the interpretability of SOP semantically using **(RQ5)** *semantic coherence* of the groups. Finally, we validate SOP's utility on **(RQ6)** *model debugging* for unwanted behaviors in correct and incorrect predictions and **(RQ7)** *scientific discovery* in cosmology.

**Experiment Setups.** We evaluate on two vision and one language datasets: ImageNet-S (Gao et al., 2022) image classification using Vision Transformer (Dosovitskiy et al., 2021) backbone, CosmoGrid (Kacprzak et al., 2023) cosmology image regression using CNN (Matilla et al., 2020), and MultiRC (Khashabi et al., 2018) text classification using BERT (Devlin et al., 2019). We use patches as image features and tokens as text features. For baselines, we compare with other SANNs including XDNN (Hesse et al., 2021), BagNet (Brendel & Bethge, 2019), and FRESH (Jain et al., 2020). Additionally, we convert post-hoc feature attribution methods to self-attributing models by using top 20% scored features as single groups and denote with "-F" (e.g. "LIME-F"), including LIME (Ribeiro et al., 2016), SHAP (Lundberg & Lee, 2017), IG (Sundararajan et al., 2017), GC (Selvaraju et al., 2016), FG (Srinivas & Fleuret, 2019), RISE (Petsiuk et al., 2018), Archipelago (Tsang et al., 2020), MFABA (Zhu et al., 2024), AGI (Pan et al., 2021), AMPE (Zhu et al., 2023) and BCos (Böhle et al., 2022). We use a maximum of 20 groups for SOP and the same 20 forward passes for all controllable baselines (Table 2). Details for datasets, models and baselines are in Appendix C.

## 4.1 Performance

**(RQ1) How Well Does SOP Perform?** As the theory suggests, SANNs can only perform well with groups of features that align with underlying correlations. We evaluate whether the learned groups allow SOP to achieve lower errors comparing to previous SANNs which rely on fixed or limited feature groups. Table 1 shows that SOP achieves

---

[2]We use binary groups to avoid leaking information resulting in unfaithful explanations.

| Category | Method | Model-Agnostic | ImageNet-S - ViT | | CosmoGrid - CNN | | MultiRC - BERT | |
| | | | Err.↓ | IOU↑ | MSE.↓ | Pur.↑ | Err.↓ | IOU↑ |
|---|---|---|---|---|---|---|---|---|
| - | Backbone | - | $0.097 \pm 0.011$ | - | $0.009 \pm 0.001$ | - | $0.318 \pm 0.021$ | - |
| Post-hoc-Converted | LIME-F | Yes | $0.327 \pm 0.014$ | $0.360 \pm 0.012$ | $0.049 \pm 0.003$ | $0.375 \pm 0.018$ | $0.475 \pm 0.031$ | $\mathbf{0.177 \pm 0.012}$ |
| | SHAP-F | Yes | $\underline{0.306 \pm 0.027}$ | $0.391 \pm 0.011$ | $\underline{0.028 \pm 0.002}$ | $0.397 \pm 0.016$ | $0.455 \pm 0.032$ | $0.135 \pm 0.020$ |
| | IG-F | Yes | $0.581 \pm 0.013$ | $0.324 \pm 0.003$ | $0.042 \pm 0.003$ | $0.391 \pm 0.011$ | $0.485 \pm 0.027$ | $0.119 \pm 0.008$ |
| | GC-F | Yes | $0.455 \pm 0.016$ | $0.398 \pm 0.015$ | $0.036 \pm 0.002$ | $0.438 \pm 0.019$ | $0.485 \pm 0.015$ | $0.099 \pm 0.001$ |
| | FG-F | Yes | $0.448 \pm 0.024$ | $\underline{0.511 \pm 0.018}$ | $0.036 \pm 0.002$ | $\underline{0.529 \pm 0.016}$ | $0.396 \pm 0.011$ | $0.107 \pm 0.005$ |
| | RISE-F | Yes | $0.732 \pm 0.009$ | $0.131 \pm 0.009$ | $0.036 \pm 0.003$ | $0.342 \pm 0.006$ | $\mathbf{0.366 \pm 0.025}$ | $0.150 \pm 0.018$ |
| | Archi-F | Yes | $0.526 \pm 0.016$ | $0.290 \pm 0.010$ | $0.069 \pm 0.002$ | $0.487 \pm 0.004$ | $0.515 \pm 0.011$ | $0.098 \pm 0.002$ |
| | MFABA-F | Yes | $0.493 \pm 0.016$ | $0.383 \pm 0.010$ | $0.035 \pm 0.003$ | $0.498 \pm 0.014$ | $0.426 \pm 0.023$ | $0.113 \pm 0.006$ |
| | AGI-F | Yes | $0.407 \pm 0.011$ | $0.439 \pm 0.012$ | $0.040 \pm 0.002$ | $0.522 \pm 0.010$ | $0.446 \pm 0.019$ | $0.147 \pm 0.012$ |
| | AMPE-F | Yes | $0.484 \pm 0.016$ | $0.417 \pm 0.012$ | $0.037 \pm 0.002$ | $0.366 \pm 0.037$ | $0.475 \pm 0.028$ | $0.116 \pm 0.011$ |
| | BCos-F[3] | No | $0.954 \pm 0.006$ | $0.234 \pm 0.003$ | - | - | - | - |
| Self-Attributing | XDNN[3] | No | $0.871 \pm 0.007$ | $0.332 \pm 0.004$ | - | - | - | - |
| | BagNet[3] | No | $0.501 \pm 0.011$ | $0.314 \pm 0.016$ | - | - | - | - |
| | FRESH[4] | No | $0.537 \pm 0.020$ | $0.464 \pm 0.015$ | - | - | $0.386 \pm 0.039$ | $0.176 \pm 0.016$ |
| | **SOP (ours)** | Yes | $\mathbf{0.267 \pm 0.017}$ | $\mathbf{0.630 \pm 0.006}$ | $\mathbf{0.025 \pm 0.002}$ | $\mathbf{0.647 \pm 0.011}$ | $\mathbf{0.366 \pm 0.021}$ | $\underline{0.176 \pm 0.008}$ |

*Table 1.* (Main Results: Error vs. Purity) This table presents error rate/MSE and IOU/purity metrics results comparing self-explaining models on ImageNet, CosmoGrid and MultiRC. We find that SOP achieves state-of-the-art performance comparing with all 14 baselines. The best result for each metric is bolded, and the second-best is underlined. For non-model-agnostic baselines, we only include for ImageNet-S where specialized pretrained models readily exist. Details of the metrics are explained in Appendix C.

| Computation Cost | Methods |
|---|---|
| 1× forward pass | IG-F, GC-F, FG-F, BCos-F, XDNN, BagNet, FRESH |
| 20× forward passes | LIME-F, SHAP-F, RISE-F, MFABA-F, AGI-F, AMPE-F, SOP |
| $\mathcal{O}(d^2)$ forward passes | Archipelago-F (due to pairwise interaction testing) |

*Table 2.* Computation cost of different attribution methods in terms of number of forward passes. SOP uses 20x forward passes, and we use the same number of forward passes for perturbation-based baselines that we compare with.

the lowest errors and MSE for all tasks. SHAP-F is the second best on vision tasks but lags on MultiRC. No other SANN consistently performs well on all tasks, demonstrating that SOP's learnable groups do enable state-of-the-art performance across diverse settings.

### 4.2 Quantitative Measures for Interpretability

**(RQ2) Can SOP Perform Well at Different Sparsity Levels?** Sparser explanations are easier to understand for humans (Lombrozo, 2007; Poursabzi-Sangdeh et al., 2021), but the best sparsity level is often unknown at training time. SOP's group generator learns to generate groups at a specific sparsity, and we test whether it performs well across other sparsity levels without retraining. Figure 3 shows that as sparsity increases ($\geq 80\%$, keeping $\leq 20\%$ features per group), SOP's error grows more slowly than other SANNs, maintaining much lower errors at extreme sparsity. Similar trends are observed in CosmoGrid and MultiRC (Appendix C.4.2). For MultiRC, FRESH slightly outperforms SOP for untrained sparsity levels, as it is optimized for language tasks. Overall, SOP trained on one sparsity also performs well on other sparsity levels at inference time.

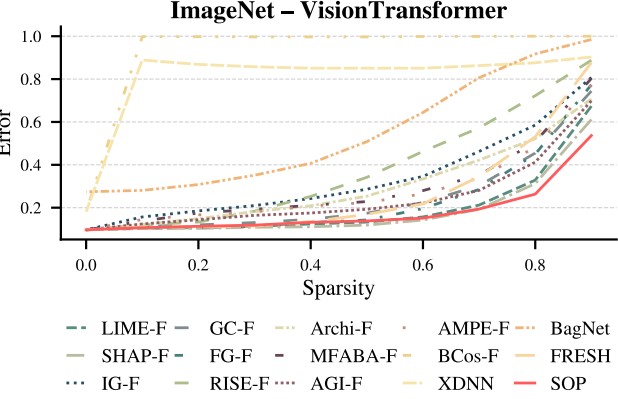

*Figure 3.* (ImageNet Sparsity vs. Error ↓) We report how error increases when sparsity increases (fewer input features are included in each group), where SOP's slowest increase is the most desired.

Beyond group size, we examine how the total number of groups affects performance. Ablation studies on ImageNet-S using 1, 2, 5, 10, 20 groups show that performance improves with more groups but saturates at 5 groups. Using only 2 groups increases the error rate by 6 percentage points compared to 5 groups, suggesting SOP can achieve

---

[3]Requires specialized architectures and thus only included for ImageNet-S where pre-trained models exist.

[4]Requires Transformer backbone and not applicable to CNN.

approximately 4x greater computational efficiency while maintaining comparable performance.

**(RQ3) How Faithful Is SOP with Respect to Classic Metrics?** While self-attributing neural networks ensure faithfulness by construction through linear aggregation, we check how well SOP performs on classic faithfulness metrics: *fidelity* (Yu & Varshney, 2017; Chen et al., 2019), *insertion*, and *deletion* (Petsiuk et al., 2018; Samek et al., 2017). Fidelity measures how well summed explanations match the model's prediction using KL-divergence, while insertion and deletion evaluate the impact of high-scoring features via perturbations and area-under-curve (AUC) computation. Definitions and results are in Appendix C.4.3, C.4.4.

All SANNs, including SOP, achieve a perfect fidelity score of 0, while no post-hoc method does (Table A5), indicating only SANNs faithfully match predictions. SOP outperforms all baselines on insertion across all tasks, and on deletion for ImageNet, while methods like Archipelago, LIME, and FRESH perform better on deletion in some cases (Tables 3 and A6). In fact, as deletion score measures how fast the predicted probability drops when the most scored features are deleted, it biases towards models that use a small number of features, regardless of how faithful the model is. SOP which relies on signals from multiple groups then naturally performs better on insertion than deletion. Ablations using smaller step sizes and occlusion values (Table A7) show consistent and thus robust results. In summary, SOP performs strong on most classic faithfulness metrics in addition to its built-in faithfulness.

**(RQ4) Do SOP's Attributions Contain Predictive Signals?** The SANN's claim of linear interpretability relies on the assumption that feature groups (which form the interpretable atoms) do not inherently encode label information. If these groups already contain predictive signals, the backbone model—not the coefficients—would drive predictions, rendering the coefficients uninformative. To validate this, we train probing models to predict labels solely from group masks: high accuracy indicates that label information is pre-encoded in the groups. Figure 4 shows that group masks from SOP achieves random accuracy (0.10%) on ImageNet-S using a CNN probing model. In comparison, probing models for groups from other SANNs achieve significantly higher accuracies, such as FG-F (13.40%) and AGI-F (10.66%). Thus SOP's generated groups do not leak information about the label compared to other SANNs, and the model's interpretablity is not weakened.

### 4.3 Semantic Measures for Interpretability

**(RQ5) How Semantically Coherent are SOP Groups?** Explanations need to be semantically coherent such as relating to object segments or human-understandable con-

| Category | Method | ImageNet | |
| --- | --- | --- | --- |
| | | Ins.↑ | Del.↓ |
| Post-Hoc-Converted | LIME-F | $0.859 \pm 0.005$ | $0.476 \pm 0.004$ |
| | SHAP-F | *0.878* $\pm 0.007$ | $0.421 \pm 0.008$ |
| | IG-F | $0.661 \pm 0.006$ | $0.664 \pm 0.008$ |
| | GC-F | $0.817 \pm 0.007$ | $0.416 \pm 0.007$ |
| | FG-F | $0.805 \pm 0.006$ | $0.430 \pm 0.004$ |
| | RISE-F | $0.635 \pm 0.007$ | $0.708 \pm 0.003$ |
| | Archi.-F | $0.719 \pm 0.004$ | $0.548 \pm 0.004$ |
| | MFABA-F | $0.720 \pm 0.005$ | $0.547 \pm 0.010$ |
| | AGI-F | $0.781 \pm 0.007$ | $0.509 \pm 0.007$ |
| | AMPE-F | $0.723 \pm 0.006$ | $0.581 \pm 0.005$ |
| | BCos-F | $0.308 \pm 0.005$ | $0.339 \pm 0.009$ |
| Self-Attributing | XDNN | $0.251 \pm 0.007$ | *0.210* $\pm 0.003$ |
| | BagNet | $0.626 \pm 0.014$ | $0.595 \pm 0.009$ |
| | FRESH | $0.759 \pm 0.003$ | $0.417 \pm 0.004$ |
| | SOP | **0.930** $\pm 0.003$ | **0.109** $\pm 0.000$ |

*Table 3.* (ImageNet Insertion/Deletion) We evaluate insertion/deletion metrics on ImageNet, and find that SOP achieves best insertion and deletion scores. This table reports percent insertion and deletion scores for ImageNet with interval of 10%. The best result for each metric is bolded, and the second-best is italicized.

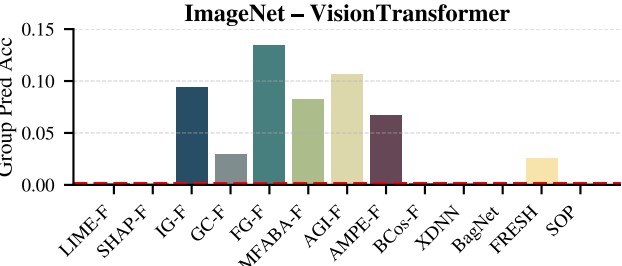

*Figure 4.* (ImageNet Group Probing Accuracy) The powerful group generator in SOP is not doing all the work and not compromising SOP's interpretability. A CNN model trained on group masks from SOP is unable to obtain accuracies more than random (0.1% accuracy), while MFABA-F, AMPE-F, IG-F etc. do. RISE and Archipelago is omitted for the significant computational cost. Results for linear and ViT probing models are in Appendix C.4.5.

cepts. We thus ask how semantically coherent are the group explanations generated by SOP compared to other models? To evaluate this, we compute *intersection-over-union* (IOU) for ImageNet-S and MultiRC where ground-truth annotations exist, and *threshold-based purity* for CosmoGrid using expert-informed metrics for cosmological structures (Matilla et al., 2020). Table 1 shows that SOP has the best semantic coherency on two vision tasks and a close second on the language task. Figure 5 illustrates an example of an image displaying a beagle, where SOP learns more semantically coherent groups than other methods without direct supervision. The exact formulations for IOU and purity, threshold ablations for CosmoGrid, and additional examples can be found in Appendix C.4.1. Additional experiments for human evaluation can be found in Appendix C.4.6.

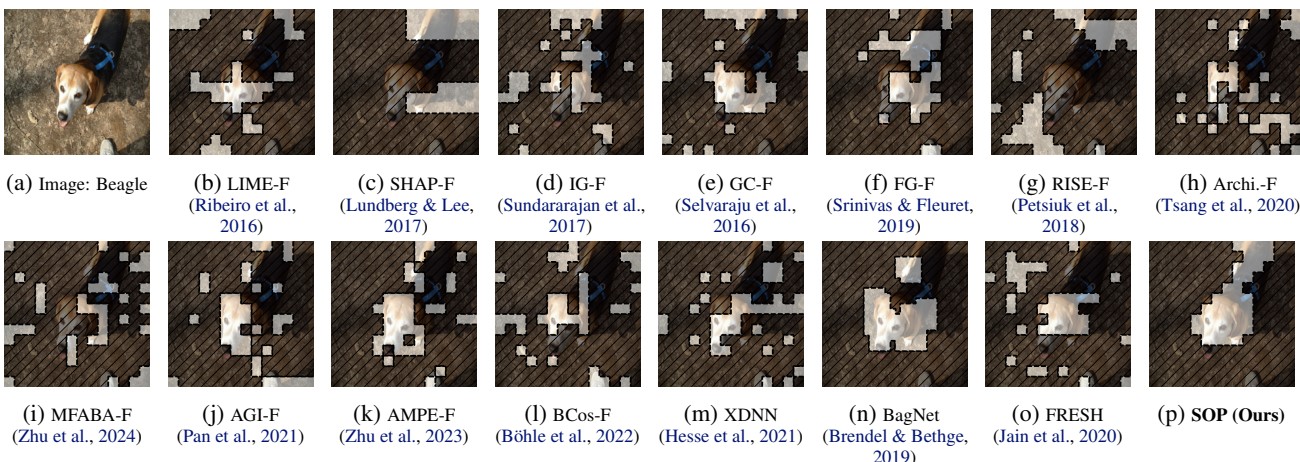

(a) Image: Beagle    (b) LIME-F (Ribeiro et al., 2016)    (c) SHAP-F (Lundberg & Lee, 2017)    (d) IG-F (Sundararajan et al., 2017)    (e) GC-F (Selvaraju et al., 2016)    (f) FG-F (Srinivas & Fleuret, 2019)    (g) RISE-F (Petsiuk et al., 2018)    (h) Archi.-F (Tsang et al., 2020)

(i) MFABA-F (Zhu et al., 2024)    (j) AGI-F (Pan et al., 2021)    (k) AMPE-F (Zhu et al., 2023)    (l) BCos-F (Böhle et al., 2022)    (m) XDNN (Hesse et al., 2021)    (n) BagNet (Brendel & Bethge, 2019)    (o) FRESH (Jain et al., 2020)    (p) **SOP (Ours)**

*Figure 5.* We show example groups from different SANNs for a Beagle in ImageNet, and find that SOP learns to generate groups more semantically coherent than other SANNs. "-F" indicates self-attributing models converted from post-hoc methods. The highlights show the groups selected by each method for ImageNet, with unused patches hatched-out. Each group has 20% features.

## 4.4 Utility of SOP Explanations

**(RQ6) Can We Use SOP Explanations to Debug Models?**
As SOP explanations directly compose the final prediction, we want to see if we can use the group explanations to identify undesirable model behaviors. One such behavior is overly relying on background features for correct predictions, indicating potential spurious correlations. We analyze SOP's explanations to check which feature groups the model use to make correct and incorrect predictions. We find that SOP uses more objects in correct (64.9%) than incorrect (57.3%) examples. Figure 6 shows similar behavior for other SANN baselines. We conjecture that using objects, instead of relying on spurious background correlations, helps the model make correct predictions. Thus, explanations from SOP and other SANNs can help illuminate the reasoning behind model behaviors.

**(RQ7) Can SOP Explanations Assist Scientific Discovery?** The ultimate goal of interpretability methods is for domain experts to use these tools and explanations in real settings. To validate the usability of our approach, we collaborate with cosmologists and use SOP to discover new knowledge about the expansion of the universe and the growth of the cosmic structure.

**Problem Formulation.** Cosmologists hope to understand the relations of cosmological structures with two cosmological parameters related to the initial state of the universe: $\Omega_m$ and $\sigma_8$. The parameter $\Omega_m$ captures the average energy density of all matter in the universe, while $\sigma_8$ describes the fluctuation of matter distribution (Abbott et al., 2022). However, these parameters are not directly measurable. What we can obtain are weak lensing mass maps, which are spatial distribution of matter density in the universe calculated using precise measurements of the shapes of $\sim 100$ million

**ImageNet-S - Vision Transformer**

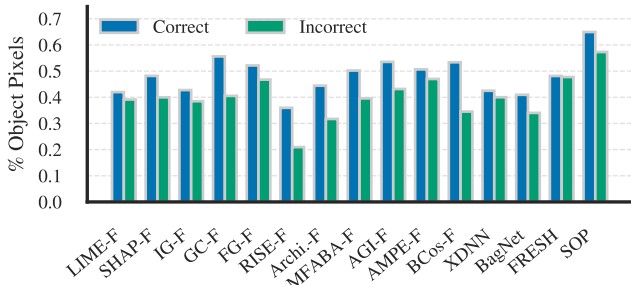

*Figure 6.* (ImageNet-S Percent of Groups that are Objects) We analyze the proportion of groups that are objects on ImageNet-S, and find that groups consistently consist more of objects when predicting correctly. This trend is consistently across methods.

galaxies (Gatti et al., 2021). While the direct relation from weak lensing maps to $\Omega_m$ and $\sigma_8$ is unknown, cosmologists create simulated weak lensing maps from different $\Omega_m$ and $\sigma_8$ values and train CNNs (Ribli et al., 2019; Matilla et al., 2020; Fluri et al., 2022) to reversely predict $\Omega_m$ and $\sigma_8$ from the mass maps. An open question in cosmology remains:

*What structures from weak lensing maps drive the inference of the cosmological parameters $\Omega_m$ and $\sigma_8$?*

As validated in Section 4.3 (RQ5), the groups from SOP correspond to more cosmological structures–voids and clusters–than other self-attributing models. Voids are large regions under-dense relative to the mean density (pixel intensity $< 0\sigma$) and appear as dark regions in the mass maps, whereas clusters are areas of concentrated high density (pixel intensity $> 3\sigma$) and appear as bright dots, as shown in Figure 7.

**Cosmological Findings.** We then use the groups automatically generated by SOP and find the following new findings related to voids/clusters and $\Omega_m/\sigma_8$:

1. Figure 8a shows that for both voids and clusters, when they are used in the prediction, they weigh higher when predicting $\Omega_m$ than $\sigma_8$.

2. Figure 8b shows that a lot of voids contribute 100% to the prediction, while most clusters contribute partially. This finding is consistent with previous work (Matilla et al., 2020) that found that voids are the most important feature in prediction for low noise maps, as when the model does use voids, it relies on them more.

Therefore, we are able to obtain explanations meaningful to expert cosmologists from SOP, which is a promising step towards meaningful scientific discovery. We include additional cosmology background in Appendix D.

## 5 Related Works

**Self-attributing neural networks** uses feature-based interpretable atoms and include per-feature models such as NAM (Agarwal et al., 2021), and group-based models such as BagNet (Brendel & Bethge, 2019) and FRESH (Jain et al., 2020). While previous works did use per-feature SANNs mainly in tabular data domains (Agarwal et al., 2021) and group-based SANNs in image and text domains (Brendel & Bethge, 2019; Jain et al., 2020), they did not formalize the problem of growing error in per-feature SANNs. Also, previous models rely on specific architectures while our framework allows the use of arbitrary pre-trained models (Li, 2023; Jo & Kim, 2023). Other self-explaining neural networks include prototype-based (Ma et al., 2024; Wen et al., 2024) and concept-based (Koh et al., 2020; Yang et al., 2023; Lai et al., 2024; Yang et al., 2024) models, which explain with prototypical examples or concepts instead of input features. Self-attributing models that we use have attributions to input features, it is not directly comparable to prototypes and concepts. Program-execution models (Lyu et al., 2023) provide faithful reasoning but do not attribute to features.

For **evaluating** SANNs, Nauta et al. (2023) advocates prioritizing performance over faithfulness as the primary metric; we adopt this approach in our evaluation. Coherence (Nauta et al., 2023) is also proposed to evaluate how well the explanation aligns with domain knowledge ground truth, using metrics such as Intersection over Union (Bau et al., 2017; Wang & Vasconcelos, 2020), outside-inside relevance ratio (Nam et al., 2020), and pointing game accuracy (Du et al., 2018; Huang & Li, 2020). We thus use IOU when there is ground truth (Gao et al., 2022; DeYoung et al., 2020) and an expert-informed threshold-based purity when the ground truth is not available (Matilla et al., 2020). Human evalua-

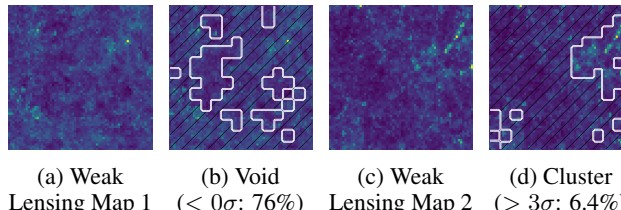

| (a) Weak Lensing Map 1 | (b) Void $(< 0\sigma: 76\%)$ | (c) Weak Lensing Map 2 | (d) Cluster $(> 3\sigma: 6.4\%)$ |

*Figure 7.* (a) and (b) show one weak lensing map with void (underdense region) found by SOP. (c) and (d) show another map with cluster (areas of concentrated high density) found.

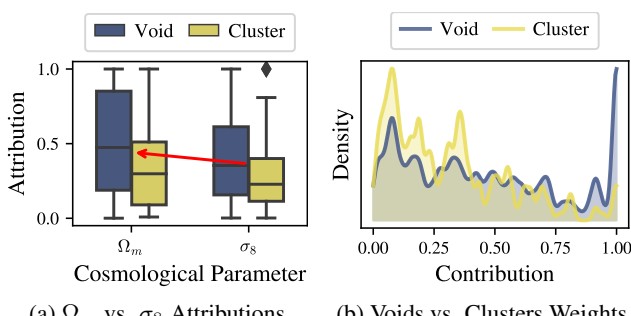

(a) $\Omega_m$ vs. $\sigma_8$ Attributions     (b) Voids vs. Clusters Weights

*Figure 8.* (a) $\Omega_m$ relies more on cosmological structures than $\sigma_8$, as shown by box plots of group weights from SOP for predicting $\Omega_m$ and $\sigma_8$. Voids on average contribute 50.7% to $\Omega_m$ and 41.5% to $\sigma_8$, and clusters contribute 34.0% to $\Omega_m$ and 28.5% to $\sigma_8$. (b) Many voids contribute 100% to the predictions, while clusters are weighted less, as shown by the density plot.

tions are used to validate utilities (Kim et al., 2022; Akula et al., 2020; Hase & Bansal, 2020). We are using a standard human distinction task from the HIVE protocol (Kim et al., 2022). Matilla et al. (2020) attempted to use post-hoc explanations in scientific discovery for cosmology. Our work uses a self-attributing model to validate the same results.

## 6 Conclusion

We propose Sum-of-Parts (SOP), a new framework for converting an arbitrary pre-trained model to a self-attributing neural network. Our framework is model-agnostic and uses end-to-end learned groups without supervision, overcoming theoretical performance limitations of per-feature or fixed-group SANNs. SOP empirically achieves state-of-the-art performance among SANNs across multiple datasets, and is shown to be useful for model debugging and uncovering insights in cosmological discovery. We hope this general framework allows people to build their own self-attributing models more easily and use the resulting explanations to extract meaningful insights from complex patterns.

## Impact Statement

This paper presents work aimed at providing interpretable explanations for machine learning models that are importantly, true to the model's reasoning process. We show that the groups of features learned in our model-agnostic framework for group-based self-attributing models provide insights in scientific discovery such as cosmology. The groups generated by our model can also be trustworthy references for decisions in other high-stakes domains such as medicine, law, and automation. We hope our work can help improve the state of explainable and trustworthy machine learning models.

## Acknowledgment

This research was partially supported by a gift from AWS AI to Penn Engineering's ASSET Center for Trustworthy AI, by ASSET Center Seed Grant, ARPA-H program on Safe and Explainable AI under the award D24AC00253-00, by NSF award CCF 2442421, and by funding from the Defense Advanced Research Projects Agency's (DARPA) SciFy program (Agreement No. HR00112520300). The views expressed are those of the author and do not reflect the official policy or position of the Department of Defense or the U.S. Government.

We thank Anton Xue, Shreya Havaldar, and everyone else at BrachioLab, and additionally Simeng Sun, Chaitanya Malaviya, Yue Yang for valuable feedback on writing. We thank Jiayi Xin for valuable discussion about this work.

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

# A    Theory Details and Proofs

In addition to the insertion error, we can define the deletion error for a SANN and a target true function when excluding a subset.

**Definition A.1.** (Deletion Error) Let $\alpha_i = \theta(x)_i h(x_{G_i})$ be the total contribution of the $i$th feature group to the prediction of a SANN. Then the *deletion error* of a self-attributing neural network $f(x) = \sum_{i=1}^{m} \theta(x)_i h(x_{G_i}) = \sum_i \alpha_i$ for a target function $f^* : \mathbb{R}^d \to \mathbb{R}$ when removing a subset of features $S$ from an input $x$ is

$$\mathrm{DelErr}(G, \alpha, S) = \left| f^*(x) - f^*(x_{\neg S}) - \sum_{S \cap G_i \neq \emptyset} \alpha_i \right|$$

$$\text{where} \;\; (x_{\neg S})_j = \begin{cases} x_j & \text{if} \;\; j \notin S \\ 0 & \text{otherwise} \end{cases}$$

Let $[d] = \{1, \dots, d\}$. Then the total deletion error over all possible deletions is $\sum_{S \subseteq [d]} \mathrm{DelErr}(G, \alpha, S)$.

We can solve the exact total deletion error for multilinear monomials with linear programs. These lowerbounds are then the minimum error in performance of SANNs attempting to learn the simple monomials.

**Theorem A.2** (Lower Bound on Deletion Error for Monomials). *Let $p : \{0,1\}^d \to \{0,1\}$ be a multilinear monomial function of $d$ variables, $p(x) = \prod_{i=1}^{d} x_i$. Then, $\sum_{S \subseteq [d]} \mathrm{DelErr}(G, \alpha, S) \geq D_{del}(\hat{\lambda})$, where $D_{del}(\hat{\lambda}) = (\hat{\lambda}_1 - \hat{\lambda}_2)^\top c$ is the lower bound, $\hat{\lambda}$ is a dual feasible point, and $c$ is a constant as defined in (9).*

*Proof.* Let $x = \mathbf{1}_d$, and let $f(x) = \sum_{i=1}^{d} \alpha_i$ with $\alpha \in \mathbb{R}^d$ be any per-feature self-attributing neural network. Consider the set of all possible perturbations to the input, or the power set of all features $\mathcal{P}([d])$, We can write the error of the self-attributing neural network under a given perturbation $S \in \mathcal{P}$ (or $S \subseteq [d]$) as

$$\mathrm{error}(\alpha, S) = \left| 1[S \neq \emptyset] - \sum_{i \in S} \alpha_i \right| = \left| c_S - M_S^\top \alpha \right| \tag{2}$$

where $(M_S, c_S)$ are defined as $(M_S)_i = \begin{cases} 1 & \text{if} \;\; i \in S \\ 0 & \text{otherwise,} \end{cases}$ and $c_S$ contains the remaining constant terms.

This captures the faithfulness notion that $\alpha_i$ is faithful if it reflects a contribution of $x_i$ to the prediction of the target function. Then, the self-attributing neural network $f(x) = \sum_{i=1}^{d} \alpha_i^*$ with $\alpha^*$ that achieves the lowest possible faithfulness error over all possible subsets is

$$\alpha^* = \arg\min_{\alpha} \sum_{S \in \mathcal{P}} \mathrm{error}(\alpha, S) \tag{3}$$

This can be more compactly written as

$$\alpha^* = \arg\min_{\alpha} \mathbf{1}^\top |c - M\alpha| \tag{4}$$

The minimum total deletion error can then be solved by the following linear program

$$P_{ins}(\alpha, \beta) = \min_{\alpha, \beta} M^\top \beta$$
$$\beta \geq c - M\alpha$$
$$\beta \geq M\alpha - c \tag{5}$$

To obtain the lower bound for total deletion error, we can solve the dual of this linear program. Given the above primal linear program, we can find the Lagrangian

$$\begin{aligned} L(\alpha, \beta, \lambda_1, \lambda_2) &= \mathbf{1}^\top \beta + \lambda_1^\top (c - M\alpha - \beta) + \lambda_2^\top (M\alpha - c - \beta) \\ &= \mathbf{1}^\top \beta - \lambda_1^\top \beta - \lambda_2^\top \beta + \lambda_1^\top (c - M\alpha) + \lambda_2^\top (M\alpha - c) \\ &= \mathbf{1}^\top \beta - (\lambda_1 + \lambda_2)^\top \beta + \lambda_1^\top c - \lambda_1^\top M\alpha + \lambda_2^\top M\alpha - \lambda_2^\top c \\ &= (\mathbf{1} - \lambda_1 - \lambda_2)^\top \beta + (\lambda_1 - \lambda_2)^\top M\alpha + (\lambda_1^\top - \lambda_2^\top) c. \end{aligned} \tag{6}$$

For the dual function to be bounded below, the coefficients of $\alpha$ and $\beta$ must be zero:

$$\frac{\partial L}{\partial \alpha} = M^\top (\lambda_2 - \lambda_1) = 0$$
$$\frac{\partial L}{\partial \beta} \mathbf{1} - \lambda_1 - \lambda_2 = 0 \Rightarrow \lambda_1 + \lambda_2 = \mathbf{1} \tag{7}$$

Since we are minimizing over $\alpha$, $\beta$, the dual objective is to maximize

$$(\lambda_1^\top - \lambda_2^\top)c$$
$$\text{subject to: } \lambda_1 + \lambda_2 = \mathbf{1}, M^\top (\lambda_2 - \lambda_1) = 0, \lambda_1, \lambda_2 \geq 0. \tag{8}$$

In summary, the dual problem is

$$D_{del}(\hat{\lambda}) = \max_{\lambda_1, \lambda_2} (\lambda_1 - \lambda_2)^\top c$$
$$\text{subject to: } \lambda_1 + \lambda_2 = \mathbf{1}, M^\top (\lambda_2 - \lambda_1) = 0, \lambda_1, \lambda_2 \geq 0. \tag{9}$$

Let $\hat{\lambda}$ be feasible, then $D_{ins}(\hat{\lambda}) = (\lambda_1 - \lambda_2)^\top c \leq \sum_{S \in \mathcal{P}} \text{InsErr}(\alpha, S)$. We can then maximize the lower bound to the primal program (5) in linear programming solvers such as CVXPY which maximizes the dual program (9).

$\square$

**Conjecture A.1** (Deletion Error for Monomials Grows Exponentially with Dimension). *Let $p : \{0,1\}^d \to \{0,1\}$ be a multilinear monomial function, $p(x) = \prod_{i=1}^{d} x_i$. Then, the lower bound of total deletion error for $p$ follows an exponential trend as dimension $d$ grows, where the lower bound is approximately $\gamma_0 + e^{\gamma_1 + \gamma_2 d}$, where $(\gamma_0, \gamma_1, \gamma_2) = (-1.030, -1.171, 0.665)$.*

We solve for $\alpha^*$ in (5) using ECOS in the `cvxpy` library for $d \in \{2, \ldots, 20\}$. To fit the exponential function, we fit a linear model to the log transform of the output which has high degree of fit (with a relative absolute error of 0.012), with the resulting exponential function shown in Figure 2b.

In other words, Theorem A.2 states that we can find the exact lower bound of the total deletion error of a monomial, and Conjecture A.1 posits that lower bound of the total deletion error of any feature attribution of a monomial grows exponentially with respect to the dimension, as visualized in Figure 2b

For high-dimensional problems, this suggests that there does not exist a feature attribution that satisfies all possible deletion tests. On the other hand, monomials can easily achieve low insertion error, as formalized in Theorem A.3.

**Theorem A.3** (Insertion Error for Monomials). *Let $p : \{0,1\}^d \to \{0,1\}$ be a multilinear monomial function of $d$ variables, $p(x) = \prod_{i=1}^{d} x_i$. Then, for all $x$, there exists a self-attributing neural network $f(x) = \sum_{i=1}^{d} \alpha_i$ for $p$ at $x$ that incurs at most 1 total insertion error.*

*Proof.* Consider $\alpha = 0_d$. If $x \neq \mathbf{1}_d$ then this achieves 0 insertion error. Otherwise, suppose $x = \mathbf{1}_d$. Then, for all subsets $S \neq [d]$, $p(x_S) = 0 = \sum_{i \in S} \alpha_i$ so $\alpha$ incurs no insertion error for all but one subset. For the last subset $S = [d]$, the insertion error is 1. Therefore, the total insertion error is at most 1 for $\alpha = 0_d$. $\square$

However, once we slightly increase the function complexity to binomials, we find that the total insertion error of any feature attribution likely follows an exponential trend with respect to $d$, as shown in Figure 2a.

**Theorem 2.3** (Lower Bound on Insertion Error for Binomials). *Let $p : \{0,1\}^d \to \{0,1,2\}$ be a multilinear binomial polynomial function. Furthermore suppose that the features can be partitioned into $(S_1, S_2, S_3)$ of equal sizes where $p(x) = \prod_{i \in S_1 \cup S_2} x_i + \prod_{j \in S_2 \cup S_3} x_j$. Then, $\sum_{S \subseteq [d]} \text{InsErr}(G, \alpha, S) \geq D_{ins}(\hat{\lambda})$, where $D_{ins}(\hat{\lambda}) = (\hat{\lambda}_1 - \hat{\lambda}_2)^\top c$ is the lower bound, $\hat{\lambda}$ is a dual feasible point, and $c$ is a constant as defined in (16).*

*Proof.* Consider $x = \mathbf{1}_d$. The addition error for a binomial function can be written as

$$\text{error}(\alpha, S) = \left| \sum_{i \in S} \alpha_i - 1[S_1 \cup S_2 \subseteq S] - 1[S_2 \cup S_3 \subseteq S] \right| = |M_S^\top \alpha - c_S| \tag{10}$$

where $(M_S, c_S)$ are defined as $(M_S)_i = \begin{cases} 1 & \text{if } i \in S \\ 0 & \text{otherwise,} \end{cases}$ and $c_S$ contains the remaining constant terms. Then, the least possible insertion error that any attribution can achieve is

$$\alpha^* = \arg\min_\alpha \sum_{S \in \mathcal{P}} \text{error}(\alpha, S) = \arg\min_\alpha \mathbf{1}^\top |c - M\alpha| \tag{11}$$

The minimum total insertion error can then be solved by the following linear program

$$\begin{aligned} P_{ins}(\alpha, \beta) = \min_{\alpha, \beta} M^\top \beta \\ \beta \geq c - M\alpha \\ \beta \geq M\alpha - c \end{aligned} \tag{12}$$

To obtain the lower bound for total insertion error, we can solve the dual of this linear program. Given the above primal linear program, we can find the Lagrangian

$$\begin{aligned} L(\alpha, \beta, \lambda_1, \lambda_2) &= \mathbf{1}^\top \beta + \lambda_1^\top (c - M\alpha - \beta) + \lambda_2^\top (M\alpha - c - \beta) \\ &= \mathbf{1}^\top \beta - \lambda_1^\top \beta - \lambda_2^\top \beta + \lambda_1^\top (c - M\alpha) + \lambda_2^\top (M\alpha - c) \\ &= \mathbf{1}^\top \beta - (\lambda_1 + \lambda_2)^\top \beta + \lambda_1^\top c - \lambda_1^\top M\alpha + \lambda_2^\top M\alpha - \lambda_2^\top c \\ &= (\mathbf{1} - \lambda_1 - \lambda_2)^\top \beta + (\lambda_1 - \lambda_2)^\top M\alpha + (\lambda_1^\top - \lambda_2^\top)c. \end{aligned} \tag{13}$$

For the dual function to be bounded below, the coefficients of $\alpha$ and $\beta$ must be zero:

$$\begin{aligned} \frac{\partial L}{\partial \alpha} &= M^\top (\lambda_2 - \lambda_1) = 0 \\ \frac{\partial L}{\partial \beta} \mathbf{1} - \lambda_1 - \lambda_2 &= 0 \Rightarrow \lambda_1 + \lambda_2 = \mathbf{1} \end{aligned} \tag{14}$$

Since we are minimizing over $\alpha$, $\beta$, the dual objective is to maximize

$$\begin{aligned} (\lambda_1^\top - \lambda_2^\top)c \\ \text{subject to: } \lambda_1 + \lambda_2 = \mathbf{1}, M^\top(\lambda_2 - \lambda_1) = 0, \lambda_1, \lambda_2 \geq 0. \end{aligned} \tag{15}$$

In summary, the dual problem is

$$\begin{aligned} D_{ins}(\hat{\lambda}) = \max_{\lambda_1, \lambda_2} (\lambda_1 - \lambda_2)^\top c \\ \text{subject to: } \lambda_1 + \lambda_2 = \mathbf{1}, M^\top(\lambda_2 - \lambda_1) = 0, \lambda_1, \lambda_2 \geq 0. \end{aligned} \tag{16}$$

Let $\hat{\lambda}$ be feasible, then $D_{ins}(\hat{\lambda}) = (\lambda_1 - \lambda_2)^\top c \leq \sum_{S \in \mathcal{P}} \text{InsErr}(\alpha, S)$. We can then maximize the lower bound to the primal program (12) in linear programming solvers such as CVXPY which maximizes the dual program (16). $\square$

**Conjecture A.2** (Insertion Error for Binomials Grows Exponentially with Dimension). *Let $p$ be a multilinear binomial function of $d$ variables as defined in Theorem 2.3. Then, the lower bound of total insertion error for $p$ follows an exponential trend as dimension $d$ grows, where the lower bound is approximately $\lambda_0 + e^{\lambda_1 + \lambda_2 d}$, where $(\lambda_0, \lambda_1, \lambda_2) = (4.778, 1.332, 0.198)$.*

We maximize the lower bound by solving the dual linear program in (16) using ECOS in the `cvxpy` library for $d \in \{2, \ldots, 20\}$. To get the exponential function, we fit a linear model to the log transform of the output, doing a grid search over the auxiliary bias term. The resulting function has a high degree of fit (with a relative absolute error of 0.188), with the resulting exponential function shown in Figure 2a.

## A.1 Groups

**Theorem A.4** (Insertion and Deletion Error for Groups). *Consider $p_1$ and $p_2$, the polynomials from Conjecture A.1 and Conjecture A.2. Then, there exists a group-based self-attributing neural network with zero deletion and insertion error for both polynomials.*

*Proof.* Let $[d]$ denote $\{1, \dots, d\}$. First let $p_1(x) = \prod_i x_i$ and consider a self-attributing neural network with one group, $f(x) = \sum_{i=1}^1 \theta(x)_i h(x_{[d]}) = \sum_{i=1}^1 1$ has one group $G = \{[d]\}$ with contribution $\alpha = \{1\}$. If $S = \emptyset$,

$$\text{DelErr}(G, \alpha, S) = \left| f^*(x) - f^*(x_{\neg S}) - \sum_{i: S \cap G_i \neq \emptyset} \alpha_i \right| = |1 - 1 - 0| = 0$$

Otherwise, no matter what subset $S$ is being tested, $S \subseteq [d]$ is always true, thus:

$$\text{DelErr}(G, \alpha, S) = \left| f^*(x) - f^*(x_{\neg S}) - \sum_{i: S \cap G_i \neq \emptyset} \alpha_i \right| = |1 - 0 - 1| = 0$$

Therefore the total grouped deletion error for $p_1$ is 0. Next let $p_2(x) = \prod_{i \in S_1 \cup S_2} x_i + \prod_{j \in S_2 \cup S_3} x_j$ and consider a self-attributing neural network with two groups, $G = \{S_1 \cup S_2, S_2 \cup S_3\}$ with contributions $\alpha = \{1, 1\}$. If $S = [d]$, then

$$\text{InsErr}(G, \alpha, S) = \left| f^*(x_S) - f^*(0) - \sum_{i: G_i \subseteq S} \alpha_i \right| = 2 - 0 - (1 + 1) = 0$$

If $S$ empty, then the insertion error is trivially 0. Otherwise suppose $S$ is missing an element from one of $S_1$ or $S_3$. WLOG suppose it is from $S_1$ but not $S_2$ or $S_3$. Then,

$$\text{InsErr}(G, \alpha, S) = \left| f^*(x_S) - f^*(0) - \sum_{i: G_i \subseteq S} \alpha_i \right| = 1 - 0 - (1) = 0$$

Otherwise, suppose we are missing elements from both $S_1$ and $S_3$. Then,

$$\text{InsErr}(G, \alpha, S) = \left| f^*(x_S) - f^*(0) - \sum_{i: G_i \subseteq S} \alpha_i \right| = 0 - 0 - (0) = 0$$

Lastly, suppose we are missing elements from $S_2$. Then,

$$\text{InsErr}(G, \alpha, S) = \left| f^*(x_S) - f^*(0) - \sum_{i: G_i \subseteq S} \alpha_i \right| = 0 - 0 = 0$$

Thus by exhaustively checking all cases, $p_2$ has zero grouped insertion error. Therefore the total grouped insertion error for $p_2$ is 0. $\qquad \square$

**Theorem A.5** (Detailed Statement: Insertion and Deletion Error for Groups for General $m$-nomial Polynomials). *Let $p : \mathbb{R}^d \to \mathbb{R}$ be any general $m$-nomial polynomial function of order $d$ with $m$ terms, $p(x) = \sum_{i=1}^m \sum_{k \in K_i} a_{ik} \prod_{j \in G_i} x_j^{b_{ijk}}$, where $(G_1, \dots, G_m) \subseteq [d]$, $G_i \neq G_{i'} \forall i, i' \in \{1, \dots, m\}$, $K_i \in \mathbb{Z}^+$ is the set of indices for terms associated with group $G_i$, and $b_{ijk} \in \mathbb{Z}^+$ is the exponent for feature $j$ in subset $i$ in the term indexed by $k \in K_i$. Then, a self-attributing neural network needs at most $m$ groups to achieve zero deletion and insertion error for polynomial $p$.*

*Proof.* Let $[d]$ denote $\{1, \dots, d\}$. Let $q_i(x) = \sum_{k \in K_i} a_{ik} \prod_{j \in G_i} x_j^{b_{ijk}}$ be the $i$th polynomial with $K_i$ terms, and then we can rewrite $p(x) = q_1(x) + \dots + q_m(x)$. We prove by induction that we can have a self-attributing neural network of $m$ groups $G = \{G_1, \dots, G_m\}$ with contributions $\alpha = \{q_1(x), \dots, q_m(x)\}$ to achieve zero deletion and insertion error for polynomial $p$. For groups and contribution scores for up to $m$ groups, we denote with $G^{(m)}$ and $\alpha^{(m)}$, and omit the superscripts when the context is clear.

**Insertion.** **Base Case:** Suppose $m = 1$, then $p_1(x) = \sum_{k \in K_1} a_{1k} \prod_{j \in G_1} x_j^{b_{1jk}} = q_1(x)$ and consider a self-attributing neural network with one group, $G^{(1)} = \{G_1\}$ with contribtuions $\alpha^{(1)} = \{q_1(x)\}$. As there are no other input features, $G_1 = [d]$. If $S$ is empty, then the insertion error is trivially 0. If $S = [d]$, then $G_1 \subseteq S$,

$$\text{InsErr}(G^{(1)}, \alpha^{(1)}, S) = \left| f^*(x_S) - f^*(0) - \sum_{i:G_i \subseteq S} \alpha_i \right| = |q_1(x) - 0 - q_1(x)| = 0$$

Otherwise, $S \subset [d] = G_1$, then $G_1 \nsubseteq S$,

$$\text{InsErr}(G^{(1)}, \alpha^{(1)}, S) = \left| f^*(x_S) - f^*(0) - \sum_{i:G_i \subseteq S} \alpha_i \right| = |0 - 0 - 0| = 0$$

We proved that we can have a self-attributing neural network with groups $G^{(1)} = \{G_1\}$ and group contributions $\alpha^{(1)} = \{q_1(x)\}$ for polynomial $p_1(x) = q_1(x)$, which only has one group. Therefore, we can need at most one group to achieve zero grouped insertion error for monomial $p_1$.

**Inductive Step:** Assume that it holds for $(m-1)$-nomial polynomial $p_{m-1}(x) = q_1(x) + \cdots + q_{m-1}(x)$ that we need at most $(m-1)$ groups to achieve zero insertion error, where the groups are $G^{(m-1)} = \{G_1, \ldots, G_{m-1}\}$ with group contributions $\alpha^{(m-1)} = \{q_1(x), \ldots, q_{m-1}(x)\}$ which means that

$$\text{InsErr}(G^{(m-1)}, \alpha^{(m-1)}, S) = \left| f^*(x_S) - f^*(0) - \sum_{i:G_i \subseteq S} \alpha_i \right| = \left| p_{m-1}(x_S) - 0 - \sum_{i:G_i \subseteq S \wedge i \neq m} q_i(x) \right| = 0 \qquad (17)$$

This holds for all $S \subseteq [d]$.

Now, we prove it for $m$-nomial polynomial $p_m(x) = q_1(x) + \cdots + q_m(x)$. There are two cases. First, if $G_m \nsubseteq S$, meaning that not all $G_m$ features are in $S$, but there are some parts of $G_m$ in $\neg S$. Then $f^*(x_S)$ does not contain the polynomial term $q_m(x)$ that uses $G_m$. Thus,

$$\text{InsErr}(G^{(m)}, \alpha^{(m)}, S) = \left| f^*(x_S) - f^*(0) - \sum_{i:G_i \subseteq S} \alpha_i \right| = \left| p_{m-1}(x_S) - 0 - \sum_{i:G_i \subseteq S \wedge i \neq m} q_i(x) \right|$$
$$= \text{InsErr}(G^{(m-1)}, \alpha^{(m-1)}, S) = 0$$

Otherwise, if $G_m \subseteq S$, meaning that all $G_m$ features are in $S$, and no features that are in $G_m$ are contained in $S$, then $f^*(x_S)$ contains the polynomial term $q_m(x)$ that uses $G_m$. Thus,

$$\text{InsErr}(G^{(m)}, \alpha^{(m)}, S) = \left| f^*(x_S) - f^*(0) - \sum_{i:G_i \subseteq S} \alpha_i \right|$$
$$= \left| (q_m(x_S) + p_{m-1}(x_S)) - 0 - \sum_{i:G_i \subseteq S} q_i(x) \right|$$
$$= \left| (q_m(x_S) + p_{m-1}(x_S)) - 0 - \left( q_m(x_S) + \sum_{i:G_i \subseteq S \wedge i \neq m} q_i(x) \right) \right|$$
$$= \left| (q_m(x_S) - q_m(x_S)) + \left( p_{m-1}(x_S) - 0 - \sum_{i:G_i \subseteq S \wedge i \neq m} q_i(x) \right) \right|$$
$$= \text{InsErr}(G^{(m-1)}, \alpha^{(m-1)}, S) = 0$$

The last steps of the above derivations use the induction from (17). Thus by exhaustively checking all cases, $p_m$ has zero group insertion error with self-attributing neural networks with groups $G^{(m)}$ and group contributions $\alpha^{(m)}$

**Deletion.** **Base Case:** Suppose $m = 1$, then $p_1(x) = \sum_{k \in K_1} a_{1k} \prod_{j \in S_1} x_j^{b_{1jk}} = q_1(x)$ and consider a self-attributing neural network with one group $G^{(1)} = \{G_1\}$ and contribution $\alpha^{(1)} = \{q_1(x)\}$. As there are no other input features, $G_1 = [d]$. If $S$ is empty, then

$$\text{DelErr}(G^{(1)}, \alpha^{(1)}, S) = \left| f^*(x) - f^*(x_{\neg S}) - \sum_{i: S \cap G_i \neq \emptyset} \alpha_i \right| = |q_1(x) - q_1(x) - 0| = 0$$

Otherwise, no matter what subset $S$ is being tested, $S \subseteq [d] = S_1$ is always true, thus $S \cap S_i \neq \emptyset$:

$$\text{DelErr}(G^{(1)}, \alpha^{(1)}, S) = \left| f^*(x) - f^*(x_{\neg S}) - \sum_{i: S \cap G_i \neq \emptyset} \alpha_i \right| = |q_1(x) - 0 - q_1(x)| = 0$$

We proved that we can have a self-attributing neural network $f(x) = \theta(x)_i h(x_{[d]}) = q_1(x)$ with groups $G^{(1)} = \{G_1\}$ and group contributions $\alpha^{(1)} = \{q_1(x)\}$ for polynomial $p_1(x) = q_1(x)$, which only has one group. Therefore, we can need at most one group to achieve zero grouped deletion error for monomial $p_1$.

**Inductive Step:** Assume that it holds for $(m-1)$-nomial polynomial $p_{m-1}(x) = q_1(x) + \cdots + q_{m-1}(x)$ that we need at most $(m-1)$ groups to achieve zero grouped deletion error, which means that

$$\text{DelErr}(G^{(m-1)}, \alpha^{(m-1)}, S) = \left| f^*(x) - f^*(x_{\neg S}) - \sum_{i: S \cap G_i \neq \emptyset} \alpha_i \right|$$

$$= \left| p_{m-1}(x) - p_{m-1}(x_{\neg S}) - \sum_{i: S \cap G_i \neq \emptyset \,\wedge\, i \neq m} q_i(x) \right| = 0 \tag{18}$$

This holds for all $S \subseteq [d]$.

Now, we prove it for $m$-nomial polynomial $p_m(x) = q_1(x) + \cdots + q_m(x)$. There are two cases. First, if $G_m \not\subseteq \neg S$, meaning that not all $G_m$ features are in $\neg S$, but there are some parts of $G_m$ in $S$. Then $f^*(x_{\neg S})$ does not contain the polynomial term $q_m(x)$ that uses $S_m$, and $S \cap G_i \neq \emptyset$. Thus,

$$\text{DelErr}(G^{(m)}, \alpha^{(m)}, S) = \left| f^*(x) - f^*(x_{\neg S}) - \sum_{i: S \cap G_i \neq \emptyset} \alpha_i \right|$$

$$= \left| \sum_{i=1}^{m} q_i(x) - p_{m-1}(x_{\neg S}) - \left( q_m(x) + \sum_{i: S \cap G_i \neq \emptyset \,\wedge\, i \neq m} q_i(x) \right) \right|$$

$$= \left| \left( q_m(x) + \sum_{i=1}^{m-1} q_i(x) \right) - p_{m-1}(x_{\neg S}) - \left( q_m(x) + \sum_{i: S \cap G_i \neq \emptyset \,\wedge\, i \neq m} q_i(x) \right) \right|$$

$$= \left| (q_m(x) + p_{m-1}(x)) - p_{m-1}(x_{\neg S}) - \left( q_m(x) + \sum_{i: S \cap G_i \neq \emptyset \,\wedge\, i \neq m} q_i(x) \right) \right|$$

$$= \left| (q_m(x) - q_m(x)) + \left( p_{m-1}(x) - p_{m-1}(x_{\neg S}) - \sum_{i: S \cap G_i \neq \emptyset \,\wedge\, i \neq m} q_i(x) \right) \right|$$

$$= \text{DelErr}(G^{(m-1)}, \alpha^{(m-1)}, S) = 0$$

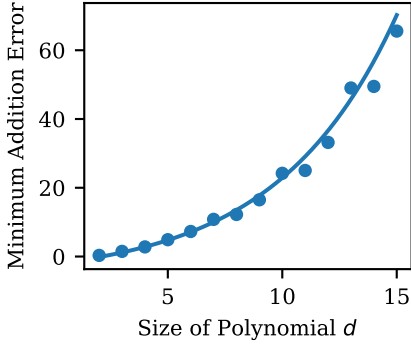

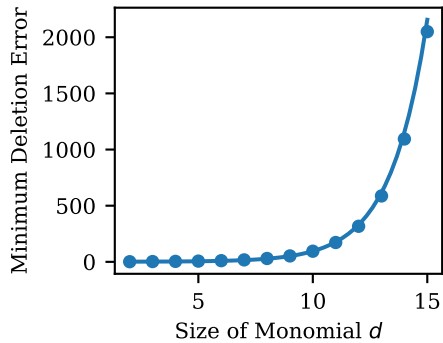

(a) Minimum total mean square insertion error for binomials.        (b) Minimum total mean square deletion error for monomials.

*Figure A9.* Errors for per-feature SANNs grow fast unavoidably. The minimum (a) total mean square insertion error of monomials of size $d$ and (b) total mean square deletion errors of binomials of size $d$ are the minima over all possible per-feature self-explaining models. The dots are the exact minima computed analytically, while the line is a best-fit exponential function.

Otherwise, if $G_m \subseteq \neg S$, meaning that all $S_m$ features are in $\neg S$, and no features that are in $G_m$ are contained in $S$, then $f^*(x_{\neg S})$ contains the polynomial term $q_m(x)$ that uses $G_m$, and $S \cap G_i = \emptyset$. Thus,

$$
\begin{aligned}
\mathrm{DelErr}(G^{(m)}, \alpha^{(m)}, S) &= \left| f^*(x) - f^*(x_{\neg S}) - \sum_{i:S\cap G_i \neq \emptyset} \alpha_i \right| \\
&= \left| \sum_{i=1}^{m} q_i(x) - (q_m(x) + p_{m-1}(x_{\neg S})) - \sum_{i:S\cap G_i \neq \emptyset \wedge i \neq m} q_i(x) \right| \\
&= \left| (q_m(x) + p_{m-1}(x)) - (q_m(x) + p_{m-1}(x_{\neg S})) - \sum_{i:S\cap G_i \neq \emptyset \wedge i \neq m} q_i(x) \right| \\
&= \left| (q_m(x) - q_m(x)) + \left( p_{m-1}(x) - p_{m-1}(x_{\neg S}) - \sum_{i:S\cap G_i \neq \emptyset \wedge i \neq m} q_i(x) \right) \right| \\
&= \mathrm{DelErr}(G^{(m-1)}, \alpha^{(m-1)}, S) = 0
\end{aligned}
$$

The last steps of the above derivations use the induction from (18). Thus by exhaustively checking all cases, $p_m$ has zero grouped deletion error with self-attributing neural network with groups $G^{(m)}$ and group contributions $\alpha^{(m)}$.

$\square$

## A.2  Alternative Mean Square Error Insertion/Deletion Errors

In the main paper, we use mean absolute error for Insertion Error and Deletion Error. Here we also compute the exact Deletion Errors for monomials and Insertion Errors for binomials up to $d = 20$. We conjecture that the two errors will also follow an exponential growth.

**Definition A.6.** (Mean Square Deletion Error) Let $\alpha_i \theta(x)_i h(x_{G_i})$ be the total contribution of the $i$th feature group to the prediction of a SANN. Then the *mean square deletion error* of a self-attributing neural network $f(x) = \sum_{i=1}^{m} \theta(x)_i h(x_{G_i}) = \sum_i \alpha_i$ for a target function $f^* : \mathbb{R}^d \to \mathbb{R}$ when removing a subset of features $S$ from an input $x$ is

$$
\mathrm{DelErr}_{\mathrm{MSE}}(G, \alpha, S) = \left\| f^*(x) - f^*(x_{\neg S}) - \sum_{S\cap G_i \neq \emptyset} \alpha_i \right\|^2
$$

$$
\text{where} \quad (x_{\neg S})_j = \begin{cases} x_j & \text{if } j \notin S \\ 0 & \text{otherwise} \end{cases}
$$

Let $[d] = \{1, \ldots, d\}$. Then the total mean square deletion error over all possible deletions is $\sum_{S \subseteq [d]} \text{DelErr}(G, \alpha, S)$.

**Theorem A.7** (Lower Bound on Mean Square Deletion Error for Monomials). *Let $p : \{0,1\}^d \to \{0,1\}$ be a multilinear monomial function of $d$ variables, $p(x) = \prod_{i=1}^{d} x_i$. Then, $\sum_{S \subseteq [d]} \text{DelErr}_{\text{MSE}}(G, \alpha, S) \geq \alpha^*$, where $\alpha^*_{del} = (M^\top M)^{-1} M^\top c$ is the lower bound, and $M$ and $c$ are constants as defined in* (28).

*Proof.* Let $x = \mathbf{1}_d$, and let $\alpha \in \mathbb{R}^d$ be any feature attribution. Consider the set of all possible perturbations to the input, or the power set of all features $\mathcal{P}$, We can write the error of the attribution under a given perturbation $S \in \mathcal{P}$ as

$$\text{error}(\alpha, S) = \left\| 1[S \neq \emptyset] - \sum_{i \in S} \alpha_i \right\|^2 = \left\| c_S - M_S^\top \alpha \right\|^2 \tag{19}$$

where $(M_S, c_S)$ are defined as $(M_S)_i = \begin{cases} 1 & \text{if } i \in S \\ 0 & \text{otherwise,} \end{cases}$ and $c_S$ contains the remaining constant terms.

This captures the faithfulness notion that $\alpha_i$ is faithful if it reflects a contribution of $\alpha_i$ to the prediction. Then, the feature attribution $\alpha^*$ that achieves the lowest possible faithfulness error over all possible subsets is

$$\alpha^* = \arg\min_\alpha \sum_{S \in \mathcal{P}} \text{error}(\alpha, S) = \arg\min_\alpha \mathbf{1}^\top \|\mathbf{c} - M\alpha\|^2 \tag{20}$$

where $M_{ij} = \begin{cases} 1 & \text{if } j \in S_i \\ 0 & \text{otherwise} \end{cases}$ for an enumeration of all elements $S_i \in \mathcal{P}$.

This is a quadratic function without constraint, and thus we can analytically solve the exact minimum by finding where the gradient is zero.

To solve

$$\min_\alpha \mathbf{1}^\top \|\mathbf{c} - M\alpha\|^2 \tag{21}$$

We first expand the squared norm as

$$\|\mathbf{c} - M\alpha\|^2 = (c - M\alpha)^\top (c - M\alpha) \tag{22}$$

Substituting into the objective function:

$$\min_\alpha \mathbf{1}^\top (c - M\alpha)^\top (c - M\alpha) \tag{23}$$

Since $\mathbf{1}^\top$ is a summation operator over all elements, this simplifies to:

$$\min_\alpha \sum_i \|c_i - (M\alpha)_i\|^2 \tag{24}$$

Then, we compute the gradient. We define

$$f(\alpha) = \sum_i \|c_i - (M\alpha)_i\|^2 \tag{25}$$

Taking the derivative with respect to $\alpha$:

$$\nabla f(\alpha) = -2M^\top (c - M\alpha) \tag{26}$$

Setting the gradient to zero:

$$M^\top M \alpha = M^\top c \tag{27}$$

Finally, we can solve for $\alpha$. If $M^\top M$ is invertible, we obtain the optimal solution:

$$\alpha^*_{del} = (M^\top M)^{-1} M^\top c. \tag{28}$$

As $M$ is the enumeration of all elements $S_i \in \mathcal{P}$, the columns in $M$ are linearly independent to each other, and thus $M \in \{0,1\}^{2^d \times d}$ is invertible with column rank $d$.

$$\text{rank}(M) = d. \tag{29}$$

By the fundamental rank theorem,

$$\text{rank}(M^\top M) = \text{rank}(M) = d \tag{30}$$

Since $M^\top M \in \mathbb{Z}^{d \times d}$, it is then invertible.

Thus we prove that we can solve the optimal solution with (28).

$\square$

**Conjecture A.3** (Mean Square Deletion Error for Monomials Grows Exponentially with Dimension). *Let $p : \{0,1\}^d \to \{0,1\}$ be a multilinear monomial function of $d \leq 20$ variables, $p(x) = \prod_{i=1}^{d} x_i$. Then, the lower bound of the total mean square deletion error for $p$ follows an exponential trend as dimension $d$ grows, where the lower bound of total mean square deletion error is approximately $\gamma_0 + e^{\gamma_1 + \gamma_2 d}$, where $(\gamma_0, \gamma_1, \gamma_2) = (-1.104, -1.720, 0.626)$.*

We solve for $\alpha^*$ in (20) using (28) for $d \in \{2, \ldots, 20\}$. To fit the exponential function, we fit a linear model to the log transform of the output which has high degree of fit (with a relative square error of -0.293), with the resulting exponential function shown in Figure A9b.

**Definition A.8.** (Mean Square Insertion Error) Let $\alpha_i = \theta(x)_i h(x_{G_i})$ be the total contribution of the $i$th feature group to the prediction of a SANN. Then, the *mean square insertion error* of a self attributing neural network $f(x) = \sum_{i=1}^{m} \theta(x)_i h(x_{G_i}) = \sum_i \alpha_i$ for a target function $f^* : \mathbb{R}^d \to \mathbb{R}$ when inserting a subset of features $S$ from an input $x$ is

$$\text{InsErr}_{\text{MSE}}(G, \alpha, S) = \left\| f^*(x_S) - f^*(0_d) - \sum_{G_i \subseteq S} \alpha_i \right\|^2$$

$$\text{where } (x_S)_j = \begin{cases} x_j & \text{if } j \in S \\ 0 & \text{otherwise} \end{cases}$$

The total mean square insertion error over all possible insertions is $\sum_{S \subseteq [d]} \text{InsErr}(G, \alpha, S)$.

**Theorem A.9** (Lower Bound on Mean Square Insertion Error for Binomials). *Let $p : \{0,1\}^d \to \{0,1,2\}$ be a multilinear binomial polynomial function of $d$ variables. Furthermore suppose that the features can be partitioned into $(S_1, S_2, S_3)$ of equal sizes where $p(x) = \prod_{i \in S_1 \cup S_2} x_i + \prod_{j \in S_2 \cup S_3} x_j$. Then, $\sum_{S \subseteq [d]} \text{InsErr}_{\text{MSE}}(G, \alpha, S) \geq \alpha^*$, where $\alpha_{del}^* = (M^\top M)^{-1} M^\top c$ is the lower bound, and $M$ and $c$ are constants as defined in (40).*

*Proof.* Consider $x = \mathbf{1}_d$. The addition error for a binomial function can be written as

$$\text{error}(\alpha, S) = \left\| \sum_{i \in S} \alpha_i - 1[S_1 \cup S_2 \subseteq S] - 1[S_2 \cup S_3 \subseteq S] \right\|^2 = \left\| M_S^\top \alpha - c_S \right\|^2 \tag{31}$$

where $(M_S, c_S)$ are defined as $(M_S)_i = \begin{cases} 1 & \text{if } i \in S \\ 0 & \text{otherwise,} \end{cases}$ and $c_S$ contains the remaining constant terms. Then, the least possible insertion error that any attribution can achieve is

$$\alpha^* = \arg\min_\alpha \sum_{S \in \mathcal{P}} \text{error}(\alpha, S) = \arg\min_\alpha \mathbf{1}^\top \left\| c - M\alpha \right\|^2 \tag{32}$$

where $(M, c)$ are constructed by stacking $(M_S, c_S)$ for some enumeration of $S \in \mathcal{P}$.

This is a quadratic function without constraint, and thus we can analytically solve the exact minimum by finding where the gradient is zero.

To solve

$$\min_{\alpha} \mathbf{1}^\top \|\mathbf{c} - M\alpha\|^2 \tag{33}$$

We first expand the squared norm as

$$\|\mathbf{c} - M\alpha\|^2 = (c - M\alpha)^\top (c - M\alpha) \tag{34}$$

Substituting into the objective function:

$$\min_{\alpha} \mathbf{1}^\top (c - M\alpha)^\top (c - M\alpha) \tag{35}$$

Since $\mathbf{1}^\top$ is a summation operator over all elements, this simplifies to:

$$\min_{\alpha} \sum_i \|c_i - (M\alpha)_i\|^2 \tag{36}$$

Then, we compute the gradient. We define

$$f(\alpha) = \sum_i \|c_i - (M\alpha)_i\|^2 \tag{37}$$

Taking the derivative with respect to $\alpha$:

$$\nabla f(\alpha) = -2M^\top (c - M\alpha) \tag{38}$$

Setting the gradient to zero:

$$M^\top M\alpha = M^\top c \tag{39}$$

Finally, we can solve for $\alpha$. If $M^\top M$ is invertible, we obtain the optimal solution:

$$\alpha_{ins}^* = (M^\top M)^{-1} M^\top c. \tag{40}$$

As $M$ is the enumeration of all elements $S_i \in \mathcal{P}$, the columns in $M$ are linearly independent to each other, and thus $M \in \{0,1\}^{2^d \times d}$ is invertible with column rank $d$.

$$\text{rank}(M) = d. \tag{41}$$

By the fundamental rank theorem,

$$\text{rank}(M^\top M) = \text{rank}(M) = d \tag{42}$$

Since $M^\top M \in \mathbb{Z}^{d \times d}$, it is then invertible.

Thus we prove that we can solve the optimal solution with (40).

$\square$

**Conjecture A.4** (Mean Square Insertion Error for Binomials Grows Exponentially with Dimension). *Let $p : \{0,1\}^d \to \{0,1,2\}$ be a multilinear binomial polynomial function of $d \leq 20$ variables. Furthermore suppose that the features can be partitioned into $(S_1, S_2, S_3)$ of equal sizes where $p(x) = \prod_{i \in S_1 \cup S_2} x_i + \prod_{j \in S_2 \cup S_3} x_j$. Then, the lower bound of total mean square insertion error for $p$ follows an exponential trend as dimension $d$ grows, where the lower bound is approximately $\lambda_0 + \exp(\lambda_1 + \lambda_2 d)$ total insertion error, where $(\lambda_0, \lambda_1, \lambda_2) = (6.451, 1.457, 0.192)$.*

We solve for $\alpha^*$ in (32) using (40) for $d \in \{2, \ldots, 20\}$. To fit the exponential function, we fit a linear model to the log transform of the output which has high degree of fit (with a relative square error of -0.110), with the resulting exponential function shown in Figure A9a.

Additionally, the coefficients for the most critical exponential terms $\gamma_2$ and $\lambda_2$ are very close between mean absolute error and mean square error. This shows that with both definitions of insertion/deletion errors we have consistent results.

### A.3 Discussion on Theoretical Proofs

While our theorems and that of Bilodeau et al. (2024) both present impossibility results for feature attributions, the assumptions and resulting theorem are different.

Bilodeau et al. (2024) put forth a result that says (put simply) that linear models cannot accurately capture complex models, where complexity is measured by having a large number of piece-wise linear components. Indeed, if we had shown that a linear model is not a good approximation of a highly non-linear model, then this would not be a novel contribution. This is also an unsurprising result (it is not surprising that a linear model cannot approximate a *highly non-linear* model).

However, our result paints a significantly bleaker picture: we show that a linear feature attribution is unable to model the extremely *simple* functions in our theorems. Our examples distill the problem to the fundamental issue in its purest form: correlated features. Specifically, we show feature attribution is impossible with only *one* group of correlated features. This is the **polar opposite** assumption than that of Bilodeau et al. (2024), and we argue that it is more surprising for feature attribution to be impossible for simpler functions than for complex functions.

Second, we provide not only a negative impossibility result for standard feature attributions, but also a positive result for grouped feature attributions that provides a path forward and motivates the approach in our submission. This is in contrast to Bilodeau et al. (2024), which only presents negative impossibility results in standard feature attributions without clear suggestions on where to go.

In summary, our theoretical results differ in Assumption (we assume simple functions with a single correlation whereas Bilodeau et al. (2024) assume complex functions with many piece-wise linearities) Theoretical results (we show both positive and negative results, whereas Bilodeau et al. (2024) only show negative results).

## B  Method Details

### B.1  Group Generator Details

The group generator $\Gamma : \mathbb{R}^d \to \{0,1\}^{m \times d}$ takes in an input $x \in \mathbb{R}^d$ and outputs $m$ groups $(g_1, \ldots, g_m)$, Each $g_i \in \{0,1\}^d$ is a binary mask such that if the feature $j$ is included in group $g_i$, then $g_{ij} = 1$, otherwise 0.

$$\Gamma(x) = (g_1, \ldots, g_m) \tag{43}$$

To generate these groups, we first project the input $x$ into embedding dimension $e$ with embedding function $h_e : \mathbb{R}^d \to \mathbb{R}^{d \times k}$. In our experiments, we typically use up to the second to last layer in the backbone model as the embedding function to obtain the contextualized embedding of each feature, and we finetune this copy of projection layer while keeping the original backbone model unchanged.

**Soft Multiheaded Self-Attention.**  Then, we use a self-attention mechanism $a : \mathbb{R}^{d \times e} \to [0,1]^{d \times d}$ (Vaswani et al., 2017) to parameterize a probability distributions over features.

$$a(x) = \text{softmax}\left(\alpha \cdot h_e(x) W_Q (h_e(x) W_K)^\top\right) \tag{44}$$

where $W_Q, W_K \in \mathbb{R}^k$ are learned parameters, and $\alpha$ is a temperature scaling hyperparameter. In practice, one might not need that many groups and can use $m < d$ groups, resulting in $a(x) \in [0,1]^{m \times d}$. However, the outputs of self-attention are continuous and dense. To make groups sparse, we binarize the attention by taking top $\tau = 0.2$ features for each group. Each group $g_i$ is then the binarized $a(x)$ where $g_{ij} = 1$ if $j$th feature is within top $\tau$ according to $a(x)$, and otherwise $g_{ij} = 0$.

**Input Embedding Encoder.**  In practice, the embedding function $h_e$ is initialized with parts of the backbone predictor $h$. The embedding function $h_e$ can be finetuned as part of the group generator while not changing the backbone predictor.

**Binary Groups and the Risk of Out-of-Distribution Data.**  The groups (i.e. the image patch subsets or language token subsets) are binary. However, the scores for each group are real valued. Since we are using discrete groups of features for making each prediction, the group masks need to be binarized. Nonbinary groups (i.e. close to zero but non-zero) will have information leakage from the nonzero input features.

A potential risk of blacking out pixels is that it will create Out-of-Distribution (OOD) data. However, modern Transformers are not significantly affected by this bias due to the significant data augmentations used during pretraining. For example,

Jain et al. (2022) show that ResNet suffers more from masking out tokens while ViT mostly is able to maintain its original prediction even when some parts are blacked out. In our ImageNet and MultiRC experiments, we use these transformer based models, and all baselines use the same transformer in fairness, except XDNN, BCos and BagNet which do not have Transformer counterparts. For CosmoGrid, though we are using a CNN-based model, the data form is very different from natural images, and zeroing parts of the map is not as OOD as it would be for natural images.

## B.2 Group Selector Details

**Sparse Multiheaded Cross-Attention.** The group selector $\theta : \{0, 1\}^{m \times d} \times \mathbb{R}^d \to [0, 1]^m$ then assigns a weight to each group such that a sparse subset of the groups get nonzero weights. The scores $(c_1, \ldots, c_m)$ are produced by a cross attention using the target class's weights $C_h \in \mathbb{R}^k$ as the query and all groups' hidden states $z = (h_h(g_1 \odot x), \ldots, h_h(g_m \odot x)) \in \mathbb{R}^{m \times k}$ as the key, where $h_h : \mathbb{R}^d \to \mathbb{R}^k$ is usually the same backbone as $h$ but outputs the hidden states with hidden dimension $k$:

$$\theta(\Gamma(x), x) = (c_1, \ldots, c_m) = \text{sparsemax}\left(\beta \cdot C_h W_{Q'}(z W_{K'})^\top\right) \tag{45}$$

where $W_{Q'}, W_{K'} \in \mathbb{R}^{k \times k}$ and $C_h \in \mathbb{R}^k$ are learned parameters, and $\beta$ is a temperature scaling factor.

Having scores on a sparse subset of groups helps avoid overloading human users. As the outputs of softmax are continuous and dense, we use a sparse variant, the sparsemax operator (Martins & Astudillo, 2016), to assign scores to the groups. The sparsemax operator uses a simplex projection to make the attention weights sparse and only assigns nonzero scores to a few groups.

In practice, we can initialize the query $C_h$ to a row of the weight matrix in the linear classifier of the pretrained model $h$, and $W_{Q'}$ and $W_{K'}$ to the identity matrix, since the pretrained model already learns a relation between the hidden states $z$ and class weights $C_h$.

**Hidden State Encoder.** In practice, the embedding function $h_e$, and encoder for hidden states $h_h$ are both initialized as parts of the backbone predictor $h$. The embedding function $h_e$ can be finetuned as part of the group generator, while the hidden state encoder $h_h$ is always kept frozen and computed at the same time of making the prediction with $h$. Therefore, $h_h$ does not incur additional forward passes.

## B.3 Scaling the Loss

Since the gradient cannot pass through the thresholded binary mask, we use a scale $\lambda \in [-1, 1]$ that is the difference between sums of attention weights of selected and unselected features and multiply that to the logits for optimization.

$$\lambda_i = \sum_{j=1}^d a(x)_{ij}^{\text{top } \tau} - \sum_{j=1}^d a(x)_{ij}^{\neg \text{top } \tau}, \quad i = 1, \ldots, m \tag{46}$$

The gradient-passing scales $(\lambda_1, \ldots, \lambda_m)$, the group scores $(c_1, \ldots, c_m)$, and predictions $(y_1, \ldots, y_m)$ are then combined with a weighted average to make the final prediction

$$f(x) = y = \lambda_1 c_1 y_1 + \cdots + \lambda_m c_m y_m \tag{47}$$

**Classification.** For classification, we multiply the scaler $\lambda$ on the logits. If the prediction from group $i$ is preferred, then $\lambda_i$ will also increase and thus up-weigh the selected features in the group. Conversely, if the prediction from group $i$ is not prefered, then $\lambda_i$ will decrease and thus down-weigh the selected features and up-weigh other features in the group.

$$\text{Scaled CrossEntropyLoss}(\vec{y}, y^*, \vec{c}, \vec{\lambda}) = \text{CrossEntropyLoss}(\sum_{i=1}^d \lambda_i c_i y_i, y^*) \tag{48}$$

where $y_i$ is the predicted logit from each group and $y^*$ is the ground truth label.

**Regression.** For regression, we multiply the negated scaler on the loss instead of the logits because we need to scale the loss instead of the absolute value.

$$\text{Scaled MSELoss}(\vec{y}, y^*, \vec{c}, \vec{\lambda}) = \min(\sum \sum_i c_i (y_i - y^*)(-\lambda_i))^2 \tag{49}$$

# C   Experiments Details

## C.1   Datasets

We evaluate SOP on two vision tasks and one language task. For vision, we use ImageNet for image classification and CosmoGrid for image regression. For language, we use MultiRC for reading comprehension framed as binary classification.

### C.1.1   IMAGENET AND IMAGENET-S

ImageNet (Russakovsky et al., 2015) is a standard image classification benchmark with 1000 classes.

- **Task:** Image classification.

- **Backbone Model:** We use a Vision Transformer (ViT) (Dosovitskiy et al., 2021) ("google/vit-base-patch16-224")[5] pretrained on ImageNet 21k and finetuned on ImageNet 1k. For the projection layer in the group generator, we use the second-to-last layer of this ViT model.

- **Input Features:** We use patches of size $3 \times 16 \times 16$.

- **Ground-Truth Segmentation for Purity:** We use ImageNet-S (Gao et al., 2022), which contains ground-truth object segment annotations for 919 classes of ImageNet. Purity scores are computed using a subset of the ImageNet validation set, with experiments on ImageNet-S done with one example for each class.

- **License:** ImageNet permits non-commercial use.

### C.1.2   COSMOGRID

CosmoGridV1 (Kacprzak et al., 2023) is an image regression dataset for predicting two cosmological parameters related to the initial state of the universe: $\Omega_m$ (related to energy density) and $\sigma_8$ (related to matter fluctuation) (Abbott et al., 2022). The task uses weak lensing maps, which are projected distributions of galaxy masses onto a 2D image (Gatti et al., 2021; Jeffrey et al., 2021).

The underlying data comes from the CosmoGridV1 suite, a set of cosmological N-body simulations produced with PKDGRAV3, a high-performance N-body treecode for self-gravitating astrophysical simulations. These simulations span different cosmological parameters, including $\Omega_m$ and $\sigma_8$. The output of the simulations are snapshots representing the distribution of matter particles as a function of position on the sky at different cosmic times (representing different distances from the observer). These simulation outputs are then post-processed to produce weak lensing mass maps, which are weighted, projected maps of the mass distribution that can be estimated from current weak lensing observations (e.g., (Jeffrey et al., 2021)). The data we use follows the post-processing from Jeffrey et al. (2021), and we refer to it as CosmoGrid.

- **Task:** Image regression to predict cosmological parameters $\Omega_m$ and $\sigma_8$. Input size: $(1 \times 66 \times 66)$, output size: 2.

- **Backbone Model:** We use a CNN (Matilla et al., 2020) trained on the CosmoGrid training set. We use the second to last layer's output as hidden states, which has the dimension $14 \times 14 \times 32$, meaning that the image of size $66 \times 66$ is divided into $14 \times 14$ grids, each grid with approximately $5 \times 5$ or $4 \times 4$ pixels.

- **License:** CosmoGrid (Kacprzak et al., 2023)[6] has a CC BY 4.0 DEED license.

### C.1.3   MULTIRC

MultiRC (Khashabi et al., 2018), sourced from the ERASER benchmark (DeYoung et al., 2020), is a reading comprehension dataset adapted for binary classification. Each example consists of a passage, a question, and an answer; the task is to predict if the answer is correct (*True* or *False*) based on the passage.

- **Task:** Binary classification for reading comprehension.

---

[5]https://huggingface.co/google/vit-base-patch16-224
[6]http://www.cosmogrid.ai/

| Method | Abbr | Model-Agnostic | Self-Attributing | Learnable Groups | Self-Attributing Ver. |
|---|---|---|---|---|---|
| LIME (Ribeiro et al., 2016) | LIME | ✓ | ✗ | ✗ | LIME-F |
| SHAP (Lundberg & Lee, 2017) | SHAP | ✓ | ✗ | ✗ | SHAP-F |
| IntGrad (Sundararajan et al., 2017) | IG | ✓ | ✗ | ✗ | IG-F |
| GradCAM (Selvaraju et al., 2016) | GC | ✓ | ✗ | ✗ | GC-F |
| FullGrad (Srinivas & Fleuret, 2019) | FG | ✓ | ✗ | ✗ | FG-F |
| RISE (Petsiuk et al., 2018) | RISE | ✓ | ✗ | ✗ | RISE-F |
| Archipelago (Tsang et al., 2020) | Archi. | ✓ | ✗ | ✗ | Archi.-F |
| MFABA (Zhu et al., 2024) | MFABA | ✓ | ✗ | ✗ | MFABA-F |
| AGI (Pan et al., 2021) | AGI | ✓ | ✗ | ✗ | AGI-F |
| AMPE (Zhu et al., 2023) | AMPE | ✓ | ✗ | ✗ | AMPE-F |
| BCos (Böhle et al., 2022) | BCos | ✗ | ✗ | ✗ | BCos-F |
| XDNN (Hesse et al., 2021) | XDNN | ✗ | ✓ | ✗ | XDNN |
| BagNet (Brendel & Bethge, 2019) | BagNet | ✗ | ✓ | ✗ | BagNet |
| FRESH (Jain et al., 2020) | FRESH | ✗ | ✓ | ✗ | FRESH |
| SOP (ours) | SOP | ✓ | ✓ | ✓ | SOP |

*Table A4.* Properties of all post-hoc feature attributions and self-attributing neural networks we use. SOP is the only attribution method that is both model agnostic and self-attributing and has learnable groups.

- **Input Features:** We use tokens as individual features.

- **Ground-Truth Annotations for Purity:** For text experiments, we use annotations from the ERASER benchmark (DeYoung et al., 2020) for purity computation.

## C.2  Model Training

### C.2.1  BACKBONE MODELS

We use Vision Transformer (Dosovitskiy et al., 2021) (google/vit-base-patch16-224) for the ImageNet backbone. For CosmoGrid experiments, we use postprocessed data from Jeffrey et al. (2021) and a CNN (Matilla et al., 2020) trained on the training set of the CosmoGrid backbone. We use BERT (Devlin et al., 2019) (bert-base-uncased) finetuned on MultiRC for the text backbone.

### C.2.2  SOP MODELS

For SOP model training, we use learning rate of 5e-6 for all experiments. For ImageNet-S and MultiRC, we use one head in the attention in the group generator. For CosmoGrid, we use four heads. We observe that using more heads result in more diverse groups, while the groups generated based on queries from the same head are similar. For all experiments, we select only 20 groups from all the possible groups, by choosing attentions created by queries spaced out evenly. We find that we do not need more than 20 groups for good performance. For ImageNet, we train for 1 epoch, but taking the checkpoint at 0.5 epoch as it already converges, while we train for 3 epochs for CosmoGrid and 20 epochs for MultiRC. For ImageNet, we use Gaussian blurring of kernel size 5 on the patches, and kernel size 15 on MultiRC, so that we obtain more contiguous groups. For CosmoGrid though, we do not use Gaussian blur because we do not assume that connected groups are better for weak lensing maps.

### C.2.3  COMPUTE RESOURCES

We use NVIDIA A100 with 80G memory and NVIDIA A6000 with 48G memory on our internal cluster for experiments. Typically, one training for ImageNet finish in a day with one epoch. The training for CosmoGrid finish in 2 hours and converge in one epoch. The training for MultiRC also converges in one epoch. The evaluation takes around 5 minutes for each method. There were preliminary experiments that take more time to debug and modify the architecture.

## C.3  Baselines

As we are building a new type of self-attributing neural networks, we compare with self-attributing neural networks that attribute to input features for all main experiments. The baselines we compare with are either already self-attributing neural networks, or converted from post-hoc attributions. The models that already fall under the class of self-attributing neural

networks are XDNN (Hesse et al., 2021), BagNet (Brendel & Bethge, 2019), FRESH (Jain et al., 2020) and SOP (ours). For post-hoc baselines, we construct a self-explaining version by passing thresholded post-hoc attributions into the backbone model to make predictions, following the framework of FRESH (Jain et al., 2020). We show all the methods we use in Table A4. We can see that SOP is the only method that is both model-agnostic and a self-attributing neural network.

**Self-Attributing Neural Networks.** For faithful baseline, the closest previous work we can compare with is FRESH (Jain et al., 2020). FRESH builds a two-stage system for faithful explanations for text. It takes the attention output from a pretrained transformer, and binarize it by taking top $\tau = 0.2$, select the input tokens based on the binary mask, and then train another transformer to take the selected group of input tokens and make prediction only based on the group.

In FRESH, they train a separate model to predict with the selected group, while we freeze the pretrained backbone and train the group generator and group selector end-to-end for prediction. Since we build attentions outside the backbone model, we are able to extract groups with any pretrained backbone model (such as both transformer and CNN), instead of only transformer-based models. FRESH (Jain et al., 2020) trains two separate components for group generation and the backbone. We assume that we don't change the backbone model, thus we only compare with a similar version to FRESH where we take the attention from the Vision Transformer and pass directly into the backbone model.

Although NAM (Agarwal et al., 2021) and other faithful models also have faithful attributions to individual features, they require a separately trained submodule for each feature. For example, BagNet (Brendel & Bethge, 2019) uses group attribution on fixed patches, while XDNN (Hesse et al., 2021) attributes to pixels. When there are existing trained specialized models for a dataset (e.g. ImageNet), then we compare with them. However, when there is no such trained model for a new dataset (e.g. CosmoGrid), we only compare with methods that can utilize already trained models. This is because it is infeasible to train a separate model from scratch for all different tasks.

**Post-hoc-converted Faithful Model.** We then compile a series of FRESH-like baselines, which take the attribution scores from different post-hoc methods, including attention, and take top $\tau = 0.2$ following Jain et al. (2020), and convert them into faithful explanations by making the prediction only based on these groups. The post-hoc attributions are from LIME (Ribeiro et al., 2016), SHAP (Lundberg & Lee, 2017), IntGrad (Sundararajan et al., 2017), GradCAM (Selvaraju et al., 2016), FullGrad (Srinivas & Fleuret, 2019), RISE (Petsiuk et al., 2018), Archipelago (Tsang et al., 2020), MFABA (Zhu et al., 2024), AGI (Pan et al., 2021), AMPE (Zhu et al., 2023), among which Archipelago already produces group attribution and we take the top predicted groups. We add "-F" after the name of each baseline to indicate that this is the faithful version using only the attributions for prediction, to differentiate with the original post-hoc methods. For SOP, we also take top $\tau = 0.2$ for each group.

**Other Baseline Details.** The baselines BCos, XDNN, and BagNet all depend on specific models and cannot be applied to any model backbone directly. For BCos, we use the simple_vit_b_patch16_224 model that is a Vision Transformer model with the linear transformations replaced by their B-cos transformation. For XDNN, they remove the bias turn in AlexNet, VGG16 and ResNet50. As they do not have available version for Vision Transformers, we use their trained xfixup_resnet50 model for ResNet50 for our experiments. FRESH also depends on the attention mechanism inside the Transformer architecture, requiring the model backbone to be a Transformer model, and thus we use the attention from Vision Transformer for ImageNet experiments. BagNet also depends on its specific CNN and cannot be applied to existing trained models. Therefore, we only compare with BCos, XDNN, BagNet and FRESH for ImageNet and only compare with other model-agnostic baselines for CosmoGrid experiment.

## C.4 Evaluation

### C.4.1 SEMANTIC COHERENCE

**Intersection-over-Union (IOU) for ImageNet-S. and MultiRC** For ImageNet-S (Gao et al., 2022), there are ground truth segmentations for a subset of images in ImageNet. For MultiRC (Khashabi et al., 2018), there are ground truth human annotated explanations. We measure intersection-over-union (IOU) of the group with the ground truth annotations (object for ImageNet-S and explanation for MultiRC). Purity of a group $g_i \in \{0, 1\}^d$ with respect to the ground truth group $\varphi \in \{0, 1\}^d$ is then

$$\text{Purity}_{\text{MultiRC}}(g_i, \varphi) = 1 - \frac{|g_i \cap \varphi|}{|g_i \cup \varphi|}) \tag{50}$$

**Threshold-based Purity for CosmoGrid.** In our collaboration with cosmologists, we identified two cosmological structures learned in our group attributions: voids and clusters. Voids are large regions that are under-dense relative to the mean density and appear as dark regions in the weak lensing mass maps, whereas clusters are areas of concentrated high density and appear as bright dots. As there are no ground-truth segments for voids and clusters, we use a proxy function from our collaborator cosmologists to compute how much of a void or cluster a certain group is. In this section, we describe how we extracted void and cluster labels from the group attributions.

Let $S$ be a group from SOP when making predictions for an input $x$. Previous work (Matilla et al., 2020) defined a cluster as a region with a mean intensity of greater than $+3\sigma$, where $\sigma$ is the standard deviation of the intensity for each weak lensing map. This provides a natural threshold for our groups: we can identify groups containing clusters as those whose features have a mean intensity of $+3\sigma$. Specifically, we calculate

$$\text{Intensity}(x, S) = \frac{1}{|S|} \sum_{i:S_i>0} x_i$$

Then, a group $S$ is labeled as a void if $\text{Intensity}(x, S) \geq 3\sigma$. Similarly, Matilla et al. (2020) define a void as a region with mean intensity less than $0$. Then, a group $S$ is labeled as a cluster if $\text{Intensity}(x, S) < 0$.

In consultation with our cosmologists collaborators, we refine the criteria in the main paper to not just use the mean intensity.

The alignment of a group $g_i$ to void is the percentage of pixels in the group that are below 0, given the mass $g_i^\top x$ is below 0. The alignment of a group $g_i$ to cluster is the percentage of pixels in the group that are above 3 standard deviation of $x$'s pixel values.

$$\begin{aligned}
\text{Align}_{\text{void}}(g_i, x) &= \frac{|g_i \odot x < 0|}{|g_i|} \cdot \mathbb{1}[g_i^\top x < 0], \\
\text{Align}_{\text{cluster}}(g_i, x) &= \frac{|g_i \odot x > 3\sigma(x)|}{|g_i|}
\end{aligned} \tag{51}$$

The purity is computed as the mean of void alignment and cluster alignment scores that are above thresholds $\tau_v$ and $\tau_c$ and averaged for the two predictions $\Omega_m$ and $\sigma_8$.

$$\begin{aligned}
\text{Purity}_{\text{CosmoGrid}}(g_i, x) &= \mathbb{1}[\text{Align}_{\text{void}}(g_i, x) > \tau_v] \\
&\quad + \mathbb{1}[\text{Align}_{\text{cluster}}(g_i, x) > \tau_c]
\end{aligned} \tag{52}$$

**CosmoGrid Purity Threshold Ablations.** In Table 1, we evaluate CosmoGrid purity with $\tau_v = 0.6$ and $\tau_c = 0.015$, so that we say a group is a void if more than 60% of its pixels are below 0 and the total mass is below 0, and a group is a cluster if more than 1.5% of its pixels are above $3\sigma(x)$.

We show more ablation of how changing the hyperparameters in CosmoGrid purity affects the results in Figure A10. The result is that SOP consistently has better purity and performance trade-off than other baselines across all varying levels of $\tau_c$ and $\tau_v$.

**Additional Examples** We show additional examples in Figure A11, A12, A13, A14, A15. The groups obtained by SOP are the most semantically localized and coherent, and thus easy to interpret.

### C.4.2 SPARSITY

Additional plots comparing how different sparsities in groups affect errors for CosmoGrid and MultiRC are shown in Figure A19. We can see that for CosmoGrid, SOP has the lowest MSE on average, and the best at sparsity 0.8 (keeping top 20% features) where it is trained. For MultiRC, SOP is competitive with FRESH while being better at the sparsity it is trained (sparsity 0.8).

### C.4.3 FIDELITY

Fidelity assesses if attributions sum up to be the same as the model's prediction (Nauta et al., 2023). Faithful attributions should have low fidelity for the attributions of each feature to accurately represent the contribution from the feature.

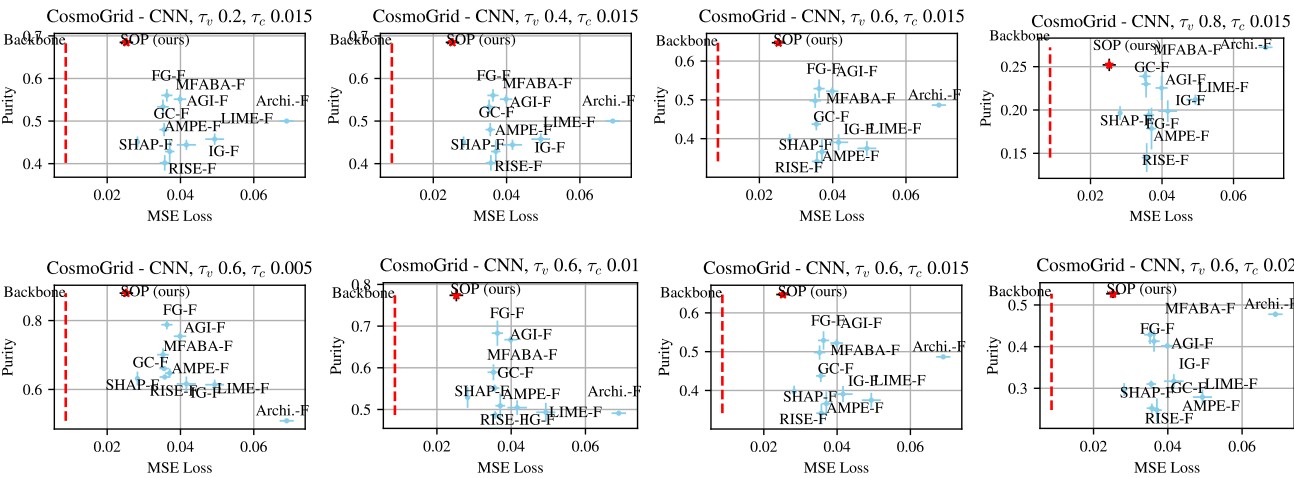

*Figure A10.* (CosmoGrid Purity Threshold Ablation) Ablation of void/cluster thresholds $\tau_v$ and $\tau_c$ for MSE Loss vs Purity. The first four plots show that when we fix the $\tau_c = 0.015$, SOP has better purity until $\tau_v = 0.8$, and the later four plots show that when we fix the $\tau_v = 0.6$, SOP consistently have the best pareto frontier than all baselines. SOP is the best on average.

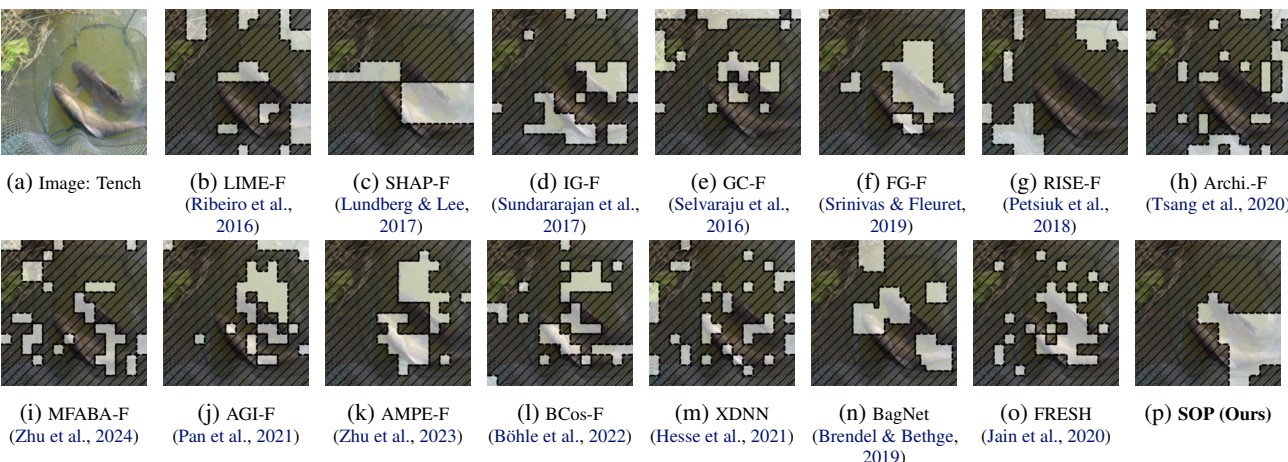

*Figure A11.* Example groups from different feature attribution methods for an image of Tench. "-F" indicates self-attributing models converted from post-hoc methods. The highlights show the groups selected by each method (top 20% attributed patches), with unused patches hatched-out.

Fidelity can be measured by the KL-Divergence between the model predicted probabilities $\hat{p}(x) \in [0,1]^k$ for all $k$ classes and the probability using summed attributions $\tilde{p}(x) \in [0,1]^k$ for all $k$ classes (Yu & Varshney, 2017; Chen et al., 2019; Anders et al., 2020).

$$\text{Fidelity} = \mathbb{E}_{x \sim \mathcal{D}}[\text{KL}(\hat{p}(x) \ || \ \tilde{p}(x))] \tag{53}$$

where $\hat{p} = \text{softmax}(f(x))$ is the predicted probability of the model, and $\tilde{p} = \text{softmax}(\sum_i \alpha_i)$ is the probability of summed attribution scores for data $x$ in distribution $\mathcal{D}$. As self-attributing neural networks all follow the design $f(x) = \sum_{i=1}^{m} \theta(x)_i h(x)_i$ and the attribution scores $\alpha_i = \theta(x)_i h(x)_i$, self-attributing neural networks by design achieve 0 fidelity. We report fidelity of post-hoc models in Table A5 and show that self-attributing neural networks are the only ones that achieve perfect fidelity.

For SOP, the sum of attribution scores is the sum of group predictions weighted by the group selector scores $\sum_i \alpha_i = \sum_i c_i y_i$. For FRESH (Jain et al., 2020), they only have one group, so the score for each class is the same as the prediction from that one group. All the self-attributing neural networks achieves fidelity of 0 by construction.

For most post-hoc baselines, the sum of attribution scores is simply the sum of all the scores for each feature. We report the fidelity scores for post-hoc methods in Table A5. We can see that no post-hoc method achieves perfect fidelity of 0.

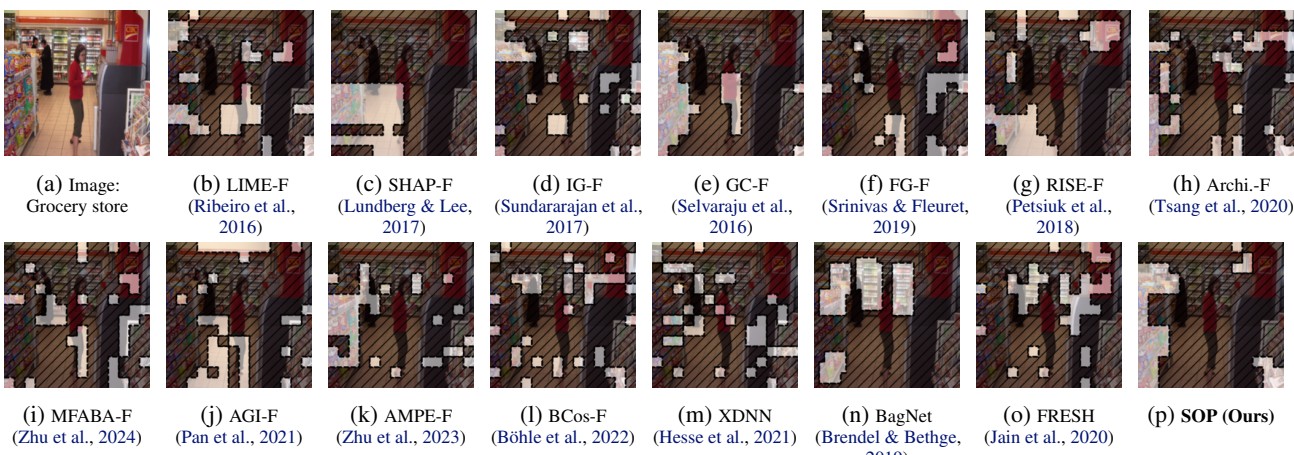

*Figure A12.* Example groups from different feature attribution methods for an image of Grocery store. "-F" indicates self-attributing models converted from post-hoc methods. The highlights show the groups selected by each method (top 20% attributed patches), with unused patches hatched-out.

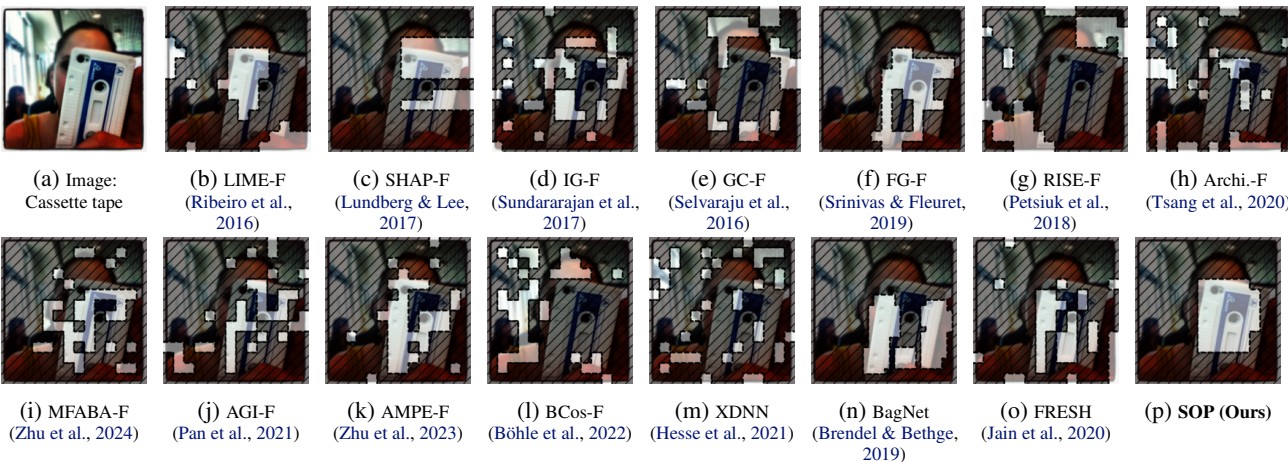

*Figure A13.* Example groups from different feature attribution methods for an image of Cassette tape. "-F" indicates self-attributing models converted from post-hoc methods. The highlights show the groups selected by each method (top 20% attributed patches), with unused patches hatched-out.

### C.4.4 INSERTION AND DELETION

**Original Insertion and Deletion.** Intuitively, if we add features from the most important to the least important one-by-one starting with a blank image, the probability for the predicted class should go up quickly. Conversely, if we delete them from the most to least important, the probability should ideally drop instantly. The mainstream insertion and deletion metrics (Samek et al., 2017; Petsiuk et al., 2018) intend to evaluate if the features are actually important by inserting or deleting them one-by-one and computing the area-under-the-curve (AUC).

As insertion and deletion criteria are designed for pixel-level instead of groups of features, we consider a modified version of insertion and deletion for groups, which grow or shrink the groups instead of adding pixels to the model. We report the percentage probability AUC instead of the absolute, to accommodate the difference in accuracy when the models are different. We use step size of 10% following previous work (Wu et al., 2024). Table 3 shows that SOP performs the best for both insertion and deletion on ImageNet. Among other methods, the perturbation-based methods such as SHAP and LIME perform better than gradient-based methods. Table A6 shows that SOP performs the best for insertion on Cosmogrid and MultiRC.

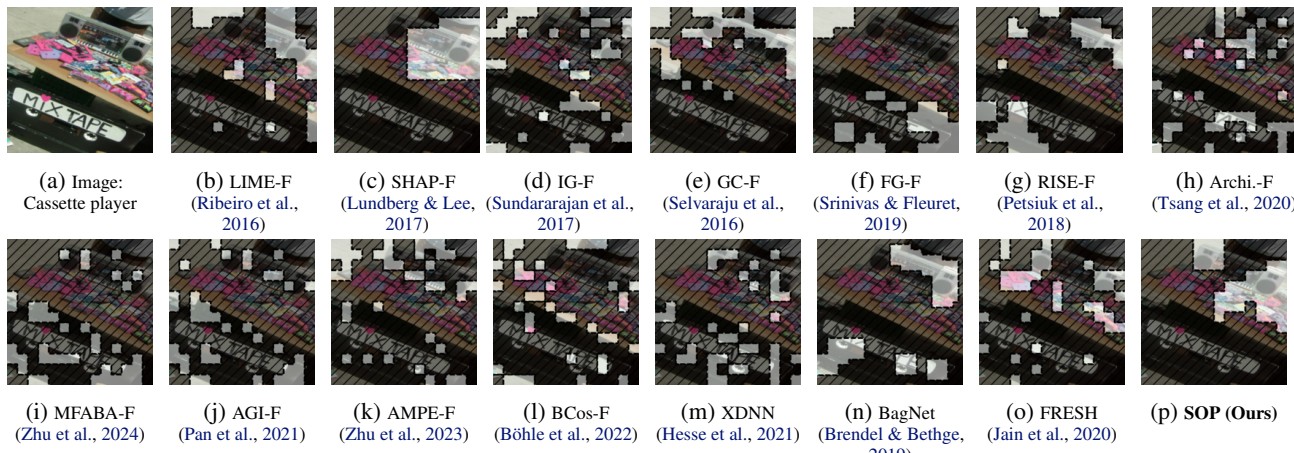

*Figure A14.* Example groups from different feature attribution methods for an image of Cassette player. "-F" indicates self-attributing models converted from post-hoc methods. The highlights show the groups selected by each method (top 20% attributed patches), with unused patches hatched-out.

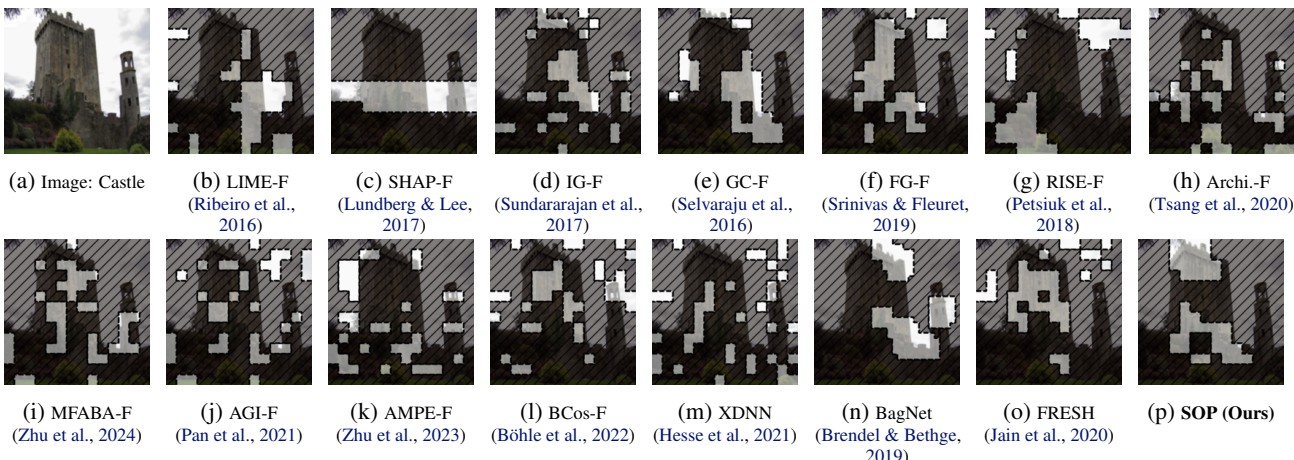

*Figure A15.* Example groups from different feature attribution methods for an image of Castle. "-F" indicates self-attributing models converted from post-hoc methods. The highlights show the groups selected by each method (top 20% attributed patches), with unused patches hatched-out.

The mainstream insertion and deletion metrics (Samek et al., 2017; Petsiuk et al., 2018) are approximations intended to evaluate the faithfulness of post-hoc methods (Nauta et al., 2023). Nevertheless, we include results for insertion and deletion for completeness.

**Adapting Insertion and Deletion for Groups.** The mainstream insertion and deletion criteria has a problem when directly used to evaluate group attribution. When inserting a single group of features, it is unclear what order one should use to insert individual features. In images, inserting pixels by row versus randomly can result in very different results. As a result, it is not immediately obvious how to directly apply the standard insertion and deletion criteria to the group setting. Nevertheless, we can consider a modified version of insertion criteria for groups, where instead of adding a single pixel at a time, we instead gradually grow the size of all groups simultaneously. This procedure is equivalent to the classic insertion criteria when there is a single group. Specifically, we grow the size of the groups gradually from with an interval of 10% of all features (approximately around 5017 pixels), and compute the AUCs of the highest predicted probability of the faithful models.

For deletion, we perform a similar procedure by shrinking the groups gradually with the same interval of 10% pixels and computing the AUCs on the highest predicted probability of the faithful models. Because each faithful model's prediction start with a different value, it is hard to compare the deletion score. Therefore, we make a small modification to the original deletion metric and penalize the cases where the prediction is wrong by setting the score of that point to 1.

| Method | ImageNet
Fidelity↓ | CosmoGrid
Fidelity↓ | MultiRC
Fidelity↓ |
|---|---|---|---|
| LIME | $3.866 \pm 0.244$ | $\mathbf{0.100 \pm 0.000}$ | $\mathbf{1.550 \pm 0.271}$ |
| SHAP | *$0.015 \pm 0.006$* | $\mathbf{0.100 \pm 0.000}$ | *$1.628 \pm 0.283$* |
| IG | $7.161 \pm 0.212$ | $\mathbf{0.100 \pm 0.000}$ | $1.921 \pm 0.276$ |
| GC | $10.406 \pm 1.098$ | $1.697 \pm 0.277$ | $1.697 \pm 0.277$ |
| FG | $13.567 \pm 0.158$ | $1.921 \pm 0.277$ | $1.921 \pm 0.277$ |
| RISE | $0.884 \pm 0.533$ | $\mathbf{0.100 \pm 0.000}$ | $2.771 \pm 0.231$ |
| Archi. | $10.850 \pm 0.354$ | $1.684 \pm 0.277$ | $1.684 \pm 0.277$ |
| MFABA | $6.674 \pm 0.166$ | $1.690 \pm 0.267$ | $1.690 \pm 0.267$ |
| AGI | $5.416 \pm 0.549$ | *$1.665 \pm 0.254$* | $1.665 \pm 0.254$ |
| AMPE | $13.671 \pm 0.326$ | $2.532 \pm 0.493$ | $2.532 \pm 0.493$ |
| BCos | $13.372 \pm 0.373$ | – | – |

*Table A5.* Fidelity scores for post-hoc methods on ImageNet, CosmoGrid, and MultiRC. Lower values indicate better fidelity. We omit numbers for self-explaining models as they by design achieve fidelity of 0. Post-hoc attribution methods are not able to achieve fidelity of 0, which empirically shows that they are not faithful to the model prediction. For CosmoGrid, to compute KL-divergence for fidelity, an additional softmax operation is applied on top of the CNN predicted logits as KL-divergence only works on probability distributions.

| Category | Method | MultiRC
Ins.↑ | MultiRC
Del.↓ | CosmoGrid
Ins.↓ | CosmoGrid
Del.↑ |
|---|---|---|---|---|---|
| Post-Hoc-Converted | LIME | $0.869 \pm 0.020$ | *$0.778 \pm 0.005$* | $0.028 \pm 0.001$ | $\underline{0.028 \pm 0.001}$ |
| | SHAP | $0.840 \pm 0.016$ | $0.839 \pm 0.017$ | $0.023 \pm 0.001$ | $\underline{0.023 \pm 0.001}$ |
| | IG | $0.878 \pm 0.009$ | $0.852 \pm 0.019$ | $0.027 \pm 0.001$ | $0.027 \pm 0.001$ |
| | GC | $0.882 \pm 0.011$ | $0.876 \pm 0.014$ | $0.026 \pm 0.001$ | $0.026 \pm 0.001$ |
| | FG | $0.928 \pm 0.009$ | $0.834 \pm 0.014$ | $0.025 \pm 0.001$ | $0.025 \pm 0.001$ |
| | RISE | *$0.961 \pm 0.017$* | $0.832 \pm 0.016$ | $0.027 \pm 0.001$ | $0.027 \pm 0.001$ |
| | Archi. | $0.669 \pm 0.021$ | $0.920 \pm 0.015$ | $0.036 \pm 0.002$ | $\mathbf{0.036 \pm 0.002}$ |
| | MFABA | $0.863 \pm 0.011$ | $0.873 \pm 0.007$ | $\underline{0.023 \pm 0.004}$ | $0.023 \pm 0.004$ |
| | AGI | $0.929 \pm 0.013$ | $0.901 \pm 0.007$ | $\underline{0.024 \pm 0.004}$ | $0.024 \pm 0.004$ |
| | AMPE | $0.868 \pm 0.007$ | $0.892 \pm 0.018$ | $0.027 \pm 0.004$ | $0.027 \pm 0.004$ |
| Self-Attributing | FRESH | $0.937 \pm 0.013$ | $\mathbf{0.710 \pm 0.028}$ | – | – |
| | SOP | $\mathbf{1.018 \pm 0.022}$ | $0.949 \pm 0.007$ | $\mathbf{0.020 \pm 0.001}$ | $0.027 \pm 0.002$ |

*Table A6.* Insertion/Deletion metrics for MultiRC and CosmoGrid. For MultiRC (left columns), higher insertion (Ins.↑) and lower deletion (Del.↓) are better. Best result is bolded, second-best is italicized. As FRESH is designed for text, it is unsurprising that it achieves better results on some metrics in text. We compute percent insertion/deletion scores as discussed in Appendix C.4.4, which is why it is possible for SOP to obtain an insertion score larger than 1. For CosmoGrid (right columns), lower insertion (Ins.↓) and higher deletion (Del.↑) are better due to MSE loss. Best result is bolded, second-best is underlined. All tables report percent insertion and deletion scores with an interval of 10%. FRESH is not applicable to CosmoGrid.

Also, as we are using different self-attributing neural networks, the base prediction can be different, making it unfair when comparing the insertion and deletion scores. We thus use the percent probability comparing with using all the input instead of the raw probability.

For self-attributing neural networks that select groups by thresholding attention, such as SOP and FRESH, we grow the size of the group according to the attention scores. For models that have fixed group sizes, such as XDNN which uses groups of single pixels, and BagNet which uses groups of fixed patches, we perform insertion/deletion by adding/removing patches from the aggregation, following the same percentage for step sizes. Similar things are performed when evaluating accuracy for different sparsity levels.

**Different Step Sizes.** Here in the Appendix, we also report more finegrained insertion and deletion scores using step sizes of $16 \times 16$ patches (which equals 256 pixels) in Table A7. The result is very similar and SOP still achieves the best insertion and deletion scores.

**Different Occlusion Strategies.** There can be different occlusion values to use when masking out the groups. To test if the choice of feature deletion values affects the results, we ran additional insertion/deletion experiments using an alternative deletion value as used in Bluecher et al. (2024) (specifically, randomly sampled color from the image). The results in

| Category | Method | ImageNet - ViT | | | |
| --- | --- | --- | --- | --- | --- |
| | | Step size 256 pix. | | Occlusion w. Colors | |
| | | Ins.↑ | Del.↓ | Ins.↑ | Del.↓ |
| Post-Hoc-Converted | LIME-F | $0.815 \pm 0.005$ | $0.428 \pm 0.004$ | $0.797 \pm 0.007$ | $0.555 \pm 0.023$ |
| | SHAP-F | $0.831 \pm 0.006$ | $0.373 \pm 0.008$ | $0.909 \pm 0.018$ | $0.524 \pm 0.023$ |
| | IG-F | $0.611 \pm 0.006$ | $0.617 \pm 0.008$ | $0.679 \pm 0.030$ | $0.786 \pm 0.059$ |
| | GC-F | $0.772 \pm 0.007$ | $0.366 \pm 0.007$ | $0.789 \pm 0.021$ | $0.540 \pm 0.031$ |
| | FG-F | $0.759 \pm 0.005$ | $0.383 \pm 0.004$ | $0.830 \pm 0.029$ | $0.447 \pm 0.026$ |
| | RISE-F | $0.590 \pm 0.008$ | $0.661 \pm 0.004$ | $0.652 \pm 0.043$ | $0.776 \pm 0.019$ |
| | Archi.-F | $0.676 \pm 0.003$ | $0.501 \pm 0.004$ | $0.824 \pm 0.058$ | $0.608 \pm 0.025$ |
| | MFABA-F | $0.674 \pm 0.006$ | $0.499 \pm 0.010$ | $0.855 \pm 0.048$ | $0.610 \pm 0.085$ |
| | AGI-F | $0.735 \pm 0.006$ | $0.462 \pm 0.008$ | $0.875 \pm 0.048$ | $0.585 \pm 0.050$ |
| | AMPE-F | $0.675 \pm 0.007$ | $0.534 \pm 0.005$ | $0.738 \pm 0.078$ | $0.645 \pm 0.041$ |
| | BCos-F | $0.257 \pm 0.005$ | $0.288 \pm 0.008$ | $0.574 \pm 0.175$ | $0.380 \pm 0.032$ |
| Self-Explaining | XDNN | $0.199 \pm 0.007$ | $0.156 \pm 0.003$ | $0.245 \pm 0.038$ | $0.254 \pm 0.044$ |
| | BagNet | $0.560 \pm 0.006$ | $0.417 \pm 0.007$ | $0.878 \pm 0.023$ | $0.228 \pm 0.022$ |
| | FRESH | $0.713 \pm 0.002$ | $0.369 \pm 0.004$ | $0.746 \pm 0.033$ | $0.512 \pm 0.046$ |
| | SOP | $\mathbf{0.890 \pm 0.004}$ | $\mathbf{0.014 \pm 0.000}$ | $\mathbf{0.910 \pm 0.010}$ | $\mathbf{0.106 \pm 0.000}$ |

*Table A7.* ImageNet Insertion (Ins.↑) and Deletion (Del.↓) metrics. Higher Ins. and lower Del. are better. Best results are **bolded**, second-best are *italicized*. Step size results use $16 \times 16$ patches (256 pixels, 5% of image). Occlusion uses a random color. All tables report percent scores with an interval of 10%.

Table A7 aligns with our previous results of replacing the deletion value with 0 and SOP still performs the best for both insertion and deletion on ImageNet. This shows that the evaluation is consistent with other feature deletion values.

**Discussion: Biases in Insertion and Deletion Tests.** SOP performs well on the insertion test for all tasks, while being only the best on deletion task for one task. We notice a bias in the deletion test. As the deletion test favors SANNs such that deleting the most important feature results in a sharp performance drop, it biases towards models that rely primarily on a small number of features. On the other hand, if a model distributes its dependence to multiple different groups of features, removing the new most important features will not lead to a large performance drop. While deletion test is initially designed to evaluate post-hoc feature attributions that attempt to explain the same backbone model, it is not well-suited to evaluate self-attributing models, as it will score models that depend on a few features more, regardless of the underlying faithfulness of the explanation.

For example, if there are multiple flowers in an image, and the model only looks at the upper left corner, while ignoring all other parts of the image, then removing the flower in the upper left corner will reduce the model predicted probability for the flower to 0. However, if the model averages the prediction from different parts of the image, its predicted probability for the flower will only drop a little bit if its most used flower is removed. This doesn't mean that the second model's explanation is any less faithful than the first one. On the other hand, the model's score for the one flower could be a smaller number that faithfully reflects the small amount of confidence reduced.

Figure A16 shows average deletion curves for SOP. We see that even as we delete features from groups, SOP is able to maintain relatively high performance, resulting in a worse deletion score. Such behavior is natural because the training objective in SOP encourages the group selector to select highly predictive groups, and multiple groups can compensate for the information missing in another group.

**Insertion/Deletion Results for CosmoGrid and MultiRC.** Table A6 shows insertion and deletion results for MultiRC and CosmoGrid. We see that SOP is consistently good on the insertion metric while LIME is better at deletion. We conjecture that this is due to different groups in SOP compensating for each other.

### C.4.5 INFORMATION LEAK FROM THE GROUP GENERATOR

We show all the results for probing if the groups contain the labels in Table A8. We can see that models trained on the SOP groups are not able to predict the labels well. While results from linear and ViT models are not differentiating different methods, CNN probing models show that the groups from other methods like FG-F and AGI-F are leaking a lot of information. Archipelago is omitted because of the significant computational cost. to generate explanations for training

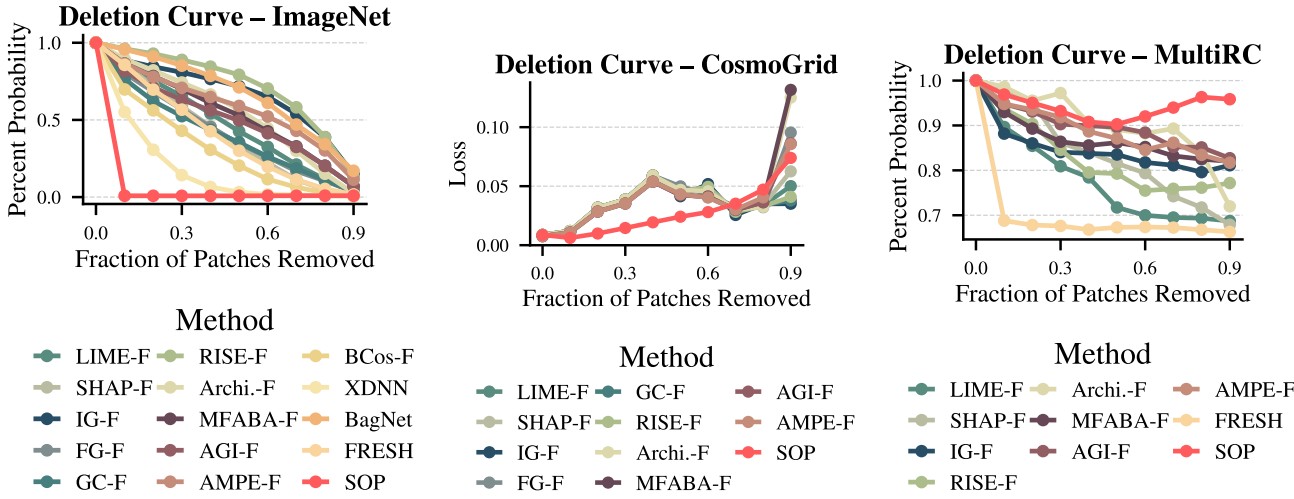

*Figure A16.* Deletion curves across datasets. SOP's multi-group approach allows it to maintain low loss when only a few input features are preserved. Other methods show unstable behavior, especially in CosmoGrid.

|  | LIME-F | SHAP-F | IG-F | GC-F | FG-F | MFABA-F | AGI-F | AMPE-F | BCos-F | XDNN | BagNet | FRESH | SOP |
|---|---|---|---|---|---|---|---|---|---|---|---|---|---|
| Model |  |  |  |  |  |  |  |  |  |  |  |  |  |
| CNN | 0.10 | 0.17 | 9.38 | 2.96 | 13.40 | 8.27 | 10.66 | 6.67 | 0.12 | 0.14 | 0.06 | 2.55 | 0.10 |
| Linear | 2.58 | 0.15 | 3.41 | 3.46 | 2.95 | 2.95 | 3.06 | 3.28 | 2.32 | 3.14 | 0.15 | 3.61 | 2.62 |
| ViT | 0.13 | 0.11 | 0.10 | 0.07 | 0.16 | 0.11 | 0.08 | 0.12 | 0.13 | 0.09 | 0.28 | 0.08 | 0.07 |

*Table A8.* (ImageNet Group Probing Model Accuracy) A model trained on group masks from SOP is unable to obtain accuracies much more than random. This indicates that the powerful group generator in SOP is not doing all the work and not compromising SOP's interpretability. RISE and Archipelago are omitted because of the significant computational cost to generate explanations for training examples.

examples, while the experiments with all other baselines already demonstrate the interpretability of linear combination in SOP.

### C.4.6 HUMAN DISTICTION TEST

SOP learns to use predictions from feature groups for the final prediction. However, if the group explanations are the same for different classes, they are not semantically relevant to the classes. We follow the HIVE protocol (Kim et al., 2022) and conduct a *human distinction task* on ImageNet-S to assess if people can predict which class the model predicts based solely on the group explanations for the classes. In the study, human evaluators are shown an image with group explanations for four classes and asked to guess the model prediction. Figure A18 shows a bar plot of human distinction task accuracies, where explanations from SOP is among the ones having most distinctive explanation for the predicted class. The large error bars do not declare a single winner, while being consistent with original experiments in HIVE (Kim et al., 2022). [7] Nevertheless, SOP groups for different classes are distinctive enough to provide meaningful explanations.

We follow HIVE (Kim et al., 2022) for constructing our distinction task for a human simulation test. Figure A18 shows the accuracy of the human distinction task and its standard deviation bootstrapped 1000 times. This human evaluation is conducted on Amazon Mechanical Turk (MTurk) on 100 examples each evaluated by 3 mturk users.

The large standard deviation is consistent with the literature and thus there is no one absolute winner method for human simulation (Kim et al., 2022). Figure A17 show the interface we display to the human evaluators.

---

[7]Here we show the attributions for different classes from the original post-hoc methods, since the converted SANNs only uses one group explanation–one for the highest predicted classes.

# Instruction

In this study, you will be asked to evaluate segments or areas in an image. An area in the image will be highlighted by black outline, and rest of the image will be hatched out. You will be asked to select the class you think is correct based on the explanation the model gives for that class.

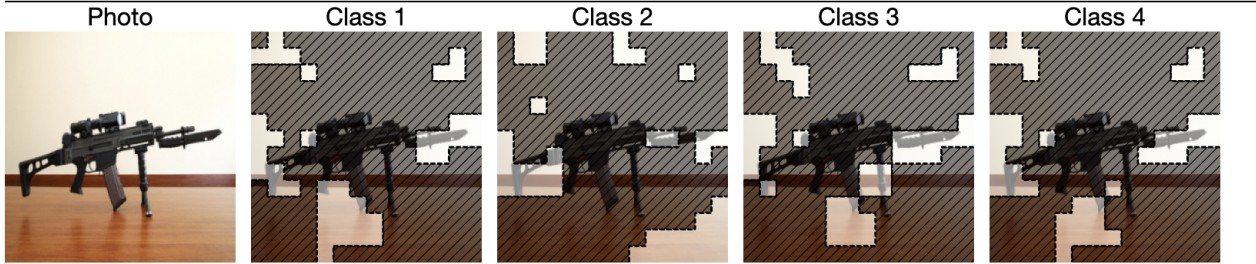

# Which class do you think is correct?

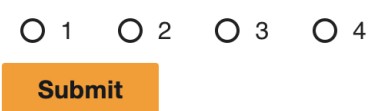

Submit

*Figure A17.* Human Distinction Task MTurk Interface. Each worker was shown the original image with four copies that have explanations for four classes highlighted. The predicted class is one of the four classes. The worker is asked to select the class that they think is correct based on the explanations for each class.

| Method | Error | IoU | Insertion | Deletion |
|---|---|---|---|---|
| FastShap-F | 0.8270 | 0.0705 | 0.4556 | 0.7974 |
| ViT-Shapley-F | 0.7323 | 0.2447 | 0.5329 | 0.5715 |
| AutoGnothi-F | 0.7095 | 0.2229 | 0.5758 | 0.5337 |

*Table A9.* (ImageNet-S) Comparison of methods by error, IoU, insertion, and deletion metrics for Additional Baselines.

We pay each worker 0.05 per task, totaling 7.5 dollars per hour. For each task, we have three workers to evaluate. We show 100 images from 100 different classes (one image per class) for each explanation.

## C.5   Computation Cost

Table 2 shows the computation cost of SOP and other models. All the baseline attribution methods whose extra number of forward passes can be controlled use at most 20 forward passes—the same number as SOP . We can see that other perturbation-based baselines also require multiple forward passes. Gradient-based methods use one forward pass and we keep them that way. For Archipelago-F, it requires pairwise comparison and thus incurs a higher cost of $O(d^2)$.

## C.6   Additional Baselines

Table A9 shows results of three additional baselines on ImageNet-S: FastShap-F (Jethani et al., 2021), ViT-Shapley-F (Covert et al., 2023), AutoGnothi-F (Wang et al., 2025). They are all post-hoc methods that attempt to approximate Shapley values faster, and we convert them as other post-hoc methods to a self-attributing version. However, as they are approximations of Shapley values, they still fall short against SOP.

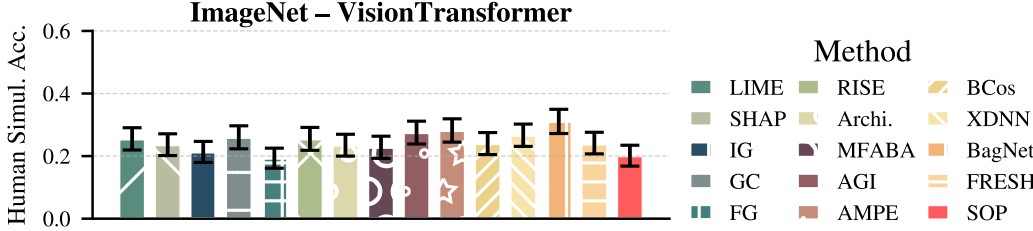

*Figure A18.* (ImageNet Human Distinction Task) We report humans distinction task accuracy for predicting which of four classes is the model prediction by looking at the attribution for each class (Kim et al., 2022). Most methods are similar in the accuracy of human predicting the correct class based solely on the explanation, and the error bars are high, which is consistent with prior work HIVE while using a larger sample size than the original HIVE.

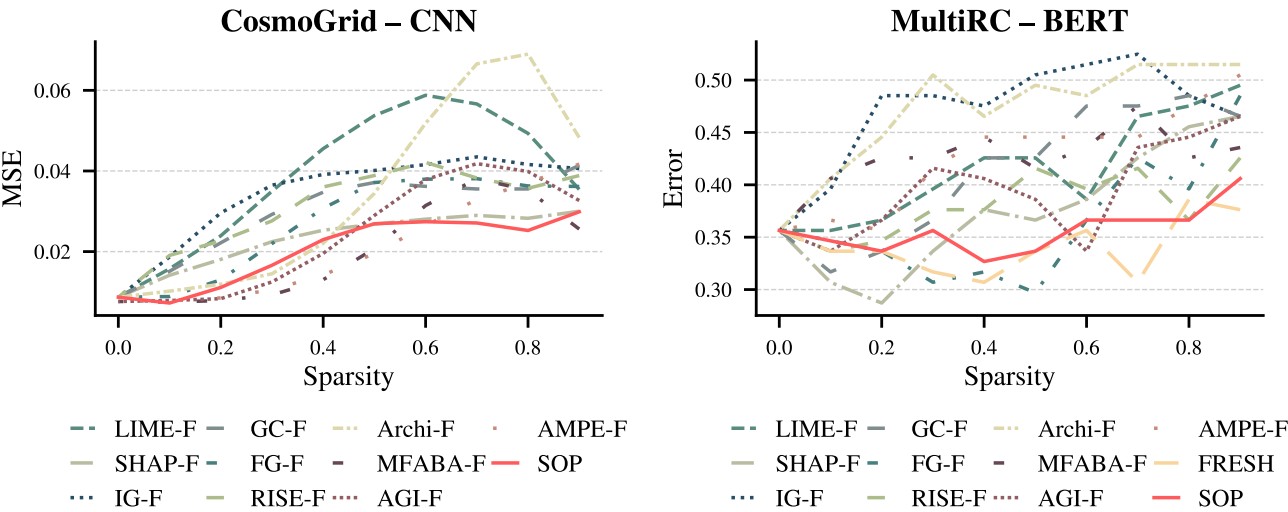

(a) (CosmoGrid Sparsity vs. MSE) We report how mean square error increases when sparsity increases (fewer input feature are included in each group), where SOP's slowest increase is the most desired.

(b) (MultiRC Sparsity vs. Error) We report how error increases when sparsity increases (fewer input feature are included in each group), where SOP and FRESH have the desired slowest increases.

*Figure A19.* (Sparsity vs Error) for CosmoGrid and MultiRC

# D   Additional Cosmology Background

While outperforming other methods on standard metrics shows the advantage of our group attributions, the ultimate goal of interpretability methods is for domain experts to use these tools and be able to use the explanations in real settings. To validate the usability of our approach, we collaborated with domain experts and used SOP to discover new cosmological knowledge about the expansion of the universe and the growth of cosmic structure. We find that the groups generated with SOP contain semantically meaningful structures to cosmologists. The resulting scores of these groups led to findings linking certain cosmological structures to the initial state of the universe, some of which were surprising and previously not known.

Weak lensing maps in cosmology calculate the spatial distribution of matter density in the universe using precise measurements of the shapes of ~100 million galaxies (Gatti et al., 2021). The shape of each galaxy is distorted (sheared and magnified) due to the curvature of spacetime induced by mass inhomogenities as light travels towards us. Cosmologists have techniques that can infer the distribution of mass in the universe from these distortions, resulting in a weak lensing map (Jeffrey et al., 2021).

**Problem Formulation.**   Cosmologists hope to use weak lensing maps to predict two key parameters related to the initial state of the universe: $\Omega_m$ and $\sigma_8$. $\Omega_m$ captures the average energy density of all matter in the universe (relative to the total energy density which includes radiation and dark energy), while $\sigma_8$ describes the fluctuation of matter distribution (see e.g. (Abbott et al., 2022)). From these parameters, a cosmologist can simulate how cosmological structures, such as galaxies,

superclusters and voids, develop throughout cosmic history. However, $\Omega_m$ and $\sigma_8$ are not directly measurable, and the inverse relation from cosmological structures in the weak lensing map to $\Omega_m$ and $\sigma_8$ is unknown.

One approach to inferring $\Omega_m$ and $\sigma_8$ from weak lensing maps, as demonstrated for example by Ribli et al. (2019); Matilla et al. (2020); Fluri et al. (2022), is to apply deep learning models that can compare measurements to simulated weak lensing maps. Even though these models have high performance, we do not fully understand how they predict $\Omega_m$ and $\sigma_8$. As a result, the following remains an open question in cosmology:

*What structures from weak lensing maps drive the inference of the cosmological parameters $\Omega_m$ and $\sigma_8$?*

In collaboration with expert cosmologists, we use convolutional networks trained to predict $\Omega_m$ and $\sigma_8$ as the backbone of an SOP model to get accurate predictions with faithful group attributions. Crucially, the guarantee of faithfulness in SOP provides confidence that the attributions reflect how the model makes its prediction, as opposed to possibly being a red herring. We then interpret and analyze these attributions and understand how structures in weak lensing maps of CosmoGridV1 (Kacprzak et al., 2023) influence $\Omega_m$ and $\sigma_8$.

It will be interesting to explore how these results change as we mimic realistic data by adding noise and measurement artifacts. Other aspects worth exploring are the role of "super-clusters" that contain multiple clusters, and how to account for the fact that voids occupy much larger areas on the sky than clusters (i.e., should we be surprised that they perform better?).

