# OpenReview forum: "Sum-of-Parts: Self-Attributing Neural Networks with End-to-End Learning of Feature Groups"
_ICML.cc/2025/Conference — ICML 2025 poster_

### Official Review · Reviewer_9FSB · 2025-03-05

**Overall Recommendation:** 4

**Summary:**

This paper proposes a new approach to producing self-explainable (attributing) NNs. It does so after identifying limitations of previous approaches. The authors thereupon introduce a group-based SANN approach. They evaluate on several datasets and investigate multiple aspects of the approach.

**Claims And Evidence:**

I think the claims are quite well supported by the experimental results, except for some which I have commented on in the other boxes.

**Essential References Not Discussed:**

I was wondering how this work relates to topics such as object factorization via approaches such as slot attention as the general idea seems quite similar and I think it would be valuable to discuss this in the context of the related works. Somewhat older, but Important references (there are most likely more novel approaches to search for):

Francesco Locatello, Dirk Weissenborn, Thomas Unterthiner, Aravindh Mahendran, Georg Heigold, Jakob Uszkoreit, Alexey Dosovitskiy, Thomas Kipf: Object-Centric Learning with Slot Attention. NeurIPS 2020

Gautam Singh, Yeongbin Kim, Sungjin Ahn: Neural Systematic Binder. ICLR 2023

This is a little ol, but might have additional references: Klaus Greff, Sjoerd van Steenkiste, Jürgen Schmidhuber: On the Binding Problem in Artificial Neural Networks. CoRR abs/2012.05208 (2020)

**Experimental Designs Or Analyses:**

There are things I can't seem to find in the experimental section. How exactly or for which task do the authors evaluate the models e.g. for RQ1? I.e. what is the task for these datasets and what does the metric MSE tell us for this task? Could the authors give more details in terms of metrics for the evaluation section?

**Methods And Evaluation Criteria:**

I have also read through the relevant parts in the supplements, but I still don't understand what the groups are based on. The authors always mention ..., but I don't understand whether these are latent features or simply partitions of the input features (e.g., image patches). Could the authors clarify this here, but also make this more explicit in the main text?

**Other Comments Or Suggestions:**

In line 201 the phrase "while the sparse number of groups ensures that the human interpretability." doesn't make sense grammatically.

Figure 3 is quite small to identify individual differences. Could the authors try enlargening this?

In line 326 the authors state "Explanations need to be semantically coherent such as relating to object segments or human-understandable concepts". I don't think this is correct. Explanations per see don't need to be this. We would hope that they do so so they are aligned with our own human knowledge, but in unsupervised learning in terms of explanations there is not guarantee that this is the case.

**Other Strengths And Weaknesses:**

I find the paper is well written and structured. The general idea seems interesting and novel.

I don't know where to write this in ICML's new template, so I'll write it here. I think the method is very cool and I actually lean towards accept, however I then noticed several things that are not optimal yet in the paper (see all of my questions and comments). And I think these need to be tackled for readers to properly follow the authors argumentations and claims. If the authors can comment and apply the suggestions throughout all of the boxes I would be very happy to raise my score to an accept.

Some weaknesses:

I find it bad to have several important, main results in the appendix. E.g., for RQ3, RQ7. In this case I would rather reduce the number of experiments in the main paper (the authors have done quite a lot, maybe move these completely to the appendix) and fill the free space with missing important results.

I am not sure about the correctness of RQ7's evaluations and what they tell us. Could the authors elaborate on this. How does it help to identify that 5% more often for correct samples the model is focussing on object partitions for stating "Thus, explanations from SOP and other SANNs can help illuminate the reasoning behind model behaviors."

There seems to be a wrong mapping to sections in line 381: "As validated in Section 4.3," in the context of RQ8, what does section 4.3 have to do with RQ8?

I also find the conclusion in line 403 a little strong given the experimental evidence. Perhaps the authors could tone this down a little or argue for why they can make this statement from the one experiment.

**Questions For Authors:**

What is the intuition why in RQ5 SOP does learn explanations that are more coherent with object segmentations? It is not trained so, so where does the influence come from?

**Relation To Broader Scientific Literature:**

I think the work relates well to prior work both on the topics of explainability and SENNs in particular.

**Theoretical Claims:**

n/a

---

> ### Author Rebuttal · Authors · 2025-04-01
>
> Thank you for the feedback! In the following response, we clarify the posed questions and commit to adjustments as needed.
>
> ## Experimental Setup Clarifications
> - *What features are groups based on?*  Groups are simply subsets of the input features (the $x_{G_i}$ from Definition 1). A SANN takes such groups and embeds them in a latent space (the $h(x_{G_i})$ from Definition 1) with a neural network to make predictions. In our experiments, $x_{G_i}$ manifests as subsets of **image patches and text tokens (line 261-262)**. We will make this more explicit by adding the following statement to Definition 1: “We refer to $x_{G_i}$, the subset of input features of $x$ corresponding to the subset $G_i$ as a group of features of $x$.”
> - *Datasets and Metrics.*  **On line 256-262, we gave the general categorizations of the tasks (image classification for ImageNet, image regression for CosmoGrid, and text classification for MultiRC), with further details in Appendix C**. To summarize: The classification tasks of ImageNet/MultiRC are to predict the right object in the image or answer the multiple choice question (Appendix C.1.1 and C.1.3). The regression task for CosmoGrid is to predict cosmological parameters of the universe from telescope images (Appendix C.1.2), where the MSE (Mean Square Error) measures how much the predicted cosmological parameters differ from true values (lower is closer and thus better).
>
> ## SOP for Model Debugging (RQ7)
> The goal of debugging is to identify a “bug”, or the reason behind errors. Here, we find that when SOP is correct, it tends to be using more of the object, whereas when SOP is incorrect, it tends to use more of the background. This analysis shows that one potential bug behind the errors of SOP is when the model looks at the background as an inaccurate proxy for the object. **This finding is only reliable because of SOP’s (and other SANNs) faithfulness guarantee–since predictions depend solely on the selected feature groups**. Without the faithfulness guarantee, one then cannot be confident if the prediction depends on the explanation. Since SOP has more semantic objects and backgrounds in its groups (from RQ5), the analysis is applicable to more SOP predictions than other SANNs.
>
> ## Clarifying the Cosmology Study (RQ8)
> **The mapping between Section 4.3 and RQ8 is correct.** Specifically, RQ8 asks a scientific question on the dependency of network predictions on cosmological structures. However, **analyzing these cosmological structures with SOP only makes sense if the SOP groups contain such cosmological structures to begin with. This fact was established previously in Section 4.3 (RQ5)**, which measured the semantic coherency of SOP groups in various settings including cosmological structures in CosmoGrid, the setting of RQ8. That is why RQ8 references Section 4.3. We will update the reference to refer specifically to RQ5.
>
> In the final conclusion, we called the result “meaningful” as the conclusion was understandable to experts, and “scientific discovery” as these are relations that experts want to understand but currently do not. **We did not intend for this to be a final irrefutable claim: rather, it is an initial step into the unknown that allows follow-up work to independently confirm or investigate more deeply.** We will adjust the wording to reflect this. These claims were made jointly with expert cosmologist collaborators, who helped us write background, justification, and conclusions of RQ8.
>
> ## Relation to Object Factorization
> At a high level, both SOP and methods such as slot attention use attention mechanisms to group input features without direct group supervision. **However, the goals (and corresponding design decisions) are quite different.**
>
> SOP forms feature groups that are beneficial for the end-to-end prediction task, without trying to reconstruct the original input. In contrast, techniques like slot attention try to break an input into its parts by learning groups that can reconstruct the input. Slot attention forces competition between groups to contain disjoint components and get coverage over the entire input, whereas SOP allows for groups to both overlap or not cover irrelevant features. Overlapping information (e.g. in Figure 1) can benefit prediction, i.e. when objects have multiple relevant contexts/groups in the image. We will add this discussion to the related work.
>
> ## Other Comments
> - We will correct the grammatical error in Line 201, and enlarge Figure 3 (a larger version can be found in Figure 4 in https://github.com/icml2025-3311/icml2025-3311/blob/main/ICML2025_3311_rebuttal.pdf ).
> - For line 326, we meant for this to refer to human understandability being improved if the explanations are semantic, and will adjust accordingly.
> - Lastly, we can certainly swap RQs to be fully in the appendix and enrich the results for other RQs in the main paper. If you can let us know which RQs should be moved, we can adjust accordingly.

---

> > ### Comment · Reviewer_9FSB · 2025-04-08
> >
> > Thanks for the detailed comments! I am happy with the clarifications and have update my rating accordingly. Regarding which RQ to move: personally I suggest to move RQ6 to the appendix as this is tied to RQ5 and I see more value in have more details of the other RQs.

---

> > > ### Author Response · Authors · 2025-04-09
> > >
> > > Thank you for your thoughtful feedback and for updating your recommendation. We also agree that moving RQ6 to the appendix would be the best. We will incorporate all the revisions into the final version. Thanks again for suggesting improvements that make our paper better!

---

### Official Review · Reviewer_cf6T · 2025-03-12

**Overall Recommendation:** 3

**Summary:**

This paper proposes a method for extending models to serve as self attributing neural networks (SANNs). The novelty is that it uses a group-based SANN where the grouping itself is learned end-to-end. SANNs are useful in that they provide an added explainable layer on top of black-box models. The paper's theoretical section shows that per-features SANN have a lower bound on the error that group-based SANN do not have. The architecture of the model is made of 3 parts: 1) a group predictor (selecting subsets of the input into groups and masking inputs), 2) a backbone model (pre-trained and frozen) applied to each masked subsets, 3) a scoring module that combines the backbone output for each group into a prediction via sparse cross-attention. The empirical section aims to show both that the method keeps the predictive power and that it provides semantically meaningful groups. On the performance, it is compared to other SANN SOTA approaches on 3 different datasets and shows it is superior to them. On the semantics of the grouping, three assessments are performed: 1) coherence (IoU and threshold-based purity are used to measure coherence and show superior behavior); 2) class-distinctiveness (this is evaluated by humans by checking if the class can be predicted by looking at the grouping); 3) semantic usefulness (do the groupings help in 2 tasks: model debugging - where model's behaviours in grouping more or less objects can be predictive of an over-reliance on background objects; cosmology-discovery assistance - where the method is shown to help understand how cosmic density is correlated to space voids).

## update after rebuttal:
I have considered the author's rebuttals of all reviews. I believe that most issue have been addressed. Therefore, I raise my rating one notch.

**Claims And Evidence:**

The claim that the method achieves better predictive performance compared to other SANN method is well supported empirically. However, the claim that the grouping provided by the method are superior to other methods is not entirely supported, as the semantic utility is not compared to other method for the cosmology experiment and has similar behaviour than other methods for the model debugging experiment.

**Essential References Not Discussed:**

It appears that the related works are properly referenced, see box above.

**Experimental Designs Or Analyses:**

The experimental design is fine and covers both accuracy preservation and semantic utility. I wish the the accuracy of the backbone itself would have been provided in the main table (1) in order to evaluate the accuracy degradation for prediction. Also some of the reported error for the baseline seem suspiciously high (> 0.7 !) for some baselines on the ImageNet task (those numbers are obtained by the authors, not lifted from the published reference).

**Methods And Evaluation Criteria:**

The evaluation criteria for accuracy are 'error' and it is not specified how it is calculated. More standard measures such as AUC and F1-score would be more meaningful. For semantic utility, IoU is used, but it is not well explained how it is calculated (using what type of annotations) and intuitively why the grouping would be more useful when intersecting with those annotations.

**Other Comments Or Suggestions:**

typos:

089-right: anddifferent

179-left: double 'in contrast'

201-right: unfinished sentence after interpretability.

393-left col: unclear explanation for (1) - missing a 'respectively'

394-right: (c) and (c)  -> (c) and (d)

1337: however + on-the-other-hand

1365: missing words after incur

1104: bad reference

**Other Strengths And Weaknesses:**

pros:
- a novel end-to-end method for SANNs that learns the grouping itself via masking.
- an interesting application to cosmological discovery where the method confirms that voids are more relevant to estimation of space density that denser clusters.

cons:
- it is unclear how the number of groups is controlled. Is it a fixed number? If yes, it should be studied how its value affects performance.
- the model and overall system is relatively complex and would benefit from pseudo-code algorithms and especially a code repository, none are given.
- given that the appendix itself has 27 (!) pages, a table of content would have been nice. Furthermore, 27 pages is too many. Please only keep what is absolutely necessary to understand the paper (first 8 pages).
- There are several typos and failed references left, please proof-read again.

**Questions For Authors:**

see above

**Relation To Broader Scientific Literature:**

It seems the paper does refer to the related literature adequately, although i'm not super familiar with this field.

**Theoretical Claims:**

The theoretical claims mostly formally proves that grouping allows for more expressibility than per-feature methods, which is not really new in my opinion and not quite center with respect to the novelty of the paper, which is more the end-to-end learnable grouping with model agnostic backbone.

---

> ### Author Rebuttal · Authors · 2025-04-01
>
> Thank you for your feedback. We respond to your questions as follows:
>
> ## Backing up the Claim that SOP’s Groups are Superior
> We would like to clarify a misunderstanding of the experiments: In referring to semantic utility, we believe the reviewer is confusing the claims on group quality with the wrong set of experiments.
>
> Specifically, **experiments showing that SOP’s groups are semantically superior are in Section 4.3,** which validate the semantic coherence of the groups. These experiments measure the semantic meaning of the groups, and evaluate all baselines on all datasets, including cosmology.
>
> We believe the reviewer is misinterpreting Section 4.4 as an evaluation of the groups. Instead, **Section 4.4 highlights how these semantic groups can be utilized in downstream tasks, not to measure groups**. In practice, practitioners will want to use and interpret the best performing model and have an easier time interpreting results that are more semantically aligned, which is SOP as validated respectively in Sections 4.2 and 4.3. For example, our cosmologist collaborators are only interested in interpreting models that can predict the ground truth parameters with low error to obtain trustworthy insights for scientific discovery.
>
>
> ## Clarification for Experiments Metrics
> - *Error Metrics.* To our knowledge, **our error metrics are within standard practice in the literature for measuring the performance in these tasks.** ImageNet is almost universally evaluated with top-1 error/accuracy. Cosmogrid uses mean-squared error as it is a regression problem, and is also consistent with prior work. Reporting F1/AUC is not common in these settings. MultiRC can be evaluated with either accuracy or F1 score, and many before us have used accuracy without a significant difference. We reported accuracy to be consistent with ImageNet, but can easily change this to F1 (the results and findings are the same).
>
> - *Exact formulation of IoU-based Semantic Utility.* **At the end of RQ5, we direct the reader to Appendix C.4.1 (line 1550) for detailed descriptions including the exact formulation for IOU, how it is calculated, and the annotations used,** which are object segmentations and human-annotated explanations. Semantic groups that align with objects in images or structures in mass maps are easier for humans to interpret in downstream analysis, as in RQ7/RQ8.
>
>
> ## Significance of Theoretical Claims
> While the idea that grouping is more expressive than per-feature methods may seem intuitive, this has remained a heuristic with no formal proof.  **Our work provides the first formal mathematical proof of this key property**. Furthermore, our theorem provides two key insights that go beyond this intuition. **First, our theorem proves that SANNs require only a finite number of groups**, contrasting other theoretical DL results that have classically assumed infinite width layers to achieve good performance. Second, our theory proves that it is **impossible for per-feature SANNs to achieve good performance.** Until now, it was not known whether per-feature SANNs could be improved with more data, larger models, or better training.
>
> ## Why do some baselines have high error?
> In our experimental setup, **we carefully controlled for all baselines to have equal compute costs and equal sparsity when measuring accuracy**. This is critical in order to attribute better performance to the SANN approach rather than more computation or more features. In Appendix C.3 we describe the common setup: all methods use at most 20 forward passes with 80% sparsity (keeping 20% features). The high error rates arise from baselines failing to be efficient with respect to compute or sparsity. RISE-F typically requires thousands of random masks to be effective, which is impractical for ImageNet inference. The performance of XDNN and BCos-F is due to their design depending on having all the features (0% sparsity) to make accurate predictions.
>
> ## How is the number of groups controlled?
> **The maximum number of groups is a hyperparameter that can be selected based on the user’s available computational resources.** We analyzed how different maximum numbers of groups affect performance on ImageNet in Figure 1 in https://github.com/icml2025-3311/icml2025-3311/blob/main/ICML2025_3311_rebuttal.pdf . We see that **performance increases with more groups until it plateaus at 5 groups**, indicating that our model can potentially be 4x more efficient!
>
> ## Request for pseudo-code and code repository
> **The requested pseudo-code and code repository are already in the submission**. The pseudo-code is in Algorithm 1 at the top of Appendix B (page 24, line 1265-1279). The code repository is included in the supplementary material.
>
>
> ## Other comments:
> - We will add the backbone performance to Table 1, which is 0.097 error for ImageNet, 0.00869 MSE for Cosmogrid, and 0.318 error for MultiRC.
> - We will add a table of contents to add structure and clarity to the Appendix.

---

### Official Review · Reviewer_rZvr · 2025-03-14

**Overall Recommendation:** 2

**Summary:**

The paper introduces Sum-of-Parts (SOP), a framework for transforming any differentiable model into a group-based Self-Attributing Neural Network (SANN). The key innovation is the end-to-end learning of feature groups without requiring explicit group supervision. SOP addresses the limitations of per-feature SANNs, which struggle with high-dimensional, correlated data, by proving that group-based SANNs can achieve zero error if the groups align with the underlying correlations in the data. The framework consists of three components: a group generator, a backbone predictor, and a group selector. SOP achieves state-of-the-art performance on vision (ImageNet, CosmoGrid) and language (MultiRC) tasks while providing interpretable and faithful explanations.

**Claims And Evidence:**

1. Claim 1: Per-feature SANNs have a lower bound on error that grows with the number of features, especially when features are highly correlated.

Evidence: The paper provides theoretical proofs (Theorems 2.3 and A.2) showing that per-feature SANNs cannot model even simple polynomial functions with correlated features. Empirical results (Figure 2) show that the error grows exponentially with the number of features.

2. Claim 2: Group-based SANNs can achieve zero error if the groups align with the underlying correlations in the data.

Evidence: The paper proves (Theorems 2.4 and A.5) that group-based SANNs can perfectly model complex polynomials with zero insertion and deletion errors. Empirical results show that SOP outperforms other SANNs and post-hoc methods on vision and language tasks.

3. SOP achieves state-of-the-art performance while maintaining interpretability.

Evidence: SOP achieves the lowest errors on ImageNet (0.267), CosmoGrid (0.025 MSE), and MultiRC (0.366 error), outperforming baselines like SHAP-F, FRESH, and BagNet (Table 1). The learned groups are validated using quantitative metrics (e.g., intersection-over-union, purity) and human evaluations.

However, the authors do not compare a recent self-interpretable works, i.e., AutoGnothi[1], which limit its trustworthiness. I suggest the authors compare SOP with AutoGnothi.

[1] Wang S, Tang H, Wang M, et al. Gnothi Seauton: Empowering Faithful Self-Interpretability in Black-Box Models[J]. ICLR, 2025.

**Essential References Not Discussed:**

[1] Wang S, Tang H, Wang M, et al. Gnothi Seauton: Empowering Faithful Self-Interpretability in Black-Box Models[J]. ICLR, 2025.

**Experimental Designs Or Analyses:**

1. Datasets: ImageNet, CosmoGrid, and MultiRC.
2. Baselines: SANNs: XDNN, BagNet, FRESH.
3. Post-hoc Methods: LIME, SHAP, IG, GradCAM, RISE, Archipelago, MFABA, AGI, AMPE, BCos.

Please compare Shapley Value based methods like KernelSHAP[2], FasSHAP[3], ViT-Shapley[4] and AutoGnothi[1].


[2] Lundberg S M, Lee S I. A unified approach to interpreting model predictions[J]. Advances in neural information processing systems, 2017, 30.
[3] Jethani N, Sudarshan M, Covert I C, et al. Fastshap: Real-time shapley value estimation[C]//International conference on learning representations. 2021.
[4]. Covert I C, Kim C, Lee S I. Learning to Estimate Shapley Values with Vision Transformers[C]//The Eleventh International Conference on Learning Representations.

**Methods And Evaluation Criteria:**

1. Methods:
- Group Generator: Uses a multi-headed self-attention mechanism to dynamically generate feature groups for each input.
- Backbone Predictor: Makes predictions for each group using a pre-trained model (e.g., Vision Transformer, CNN, BERT).
- Group Selector: Assigns scores to each group using a sparse cross-attention mechanism, ensuring that only a few groups contribute to the final prediction.

2. Evaluation Criteria:
- Performance: Error rates (ImageNet, MultiRC) and mean squared error (CosmoGrid).
- Interpretability: Intersection-over-union (IOU) for ImageNet-S, threshold-based purity for CosmoGrid, and human distinction tasks.
- Faithfulness: Fidelity (KL-divergence between model predictions and summed attributions), insertion and deletion tests.
- Utility: Model debugging (correct vs. incorrect predictions) and scientific discovery (cosmology insights).

**Other Comments Or Suggestions:**

See above comments.

**Other Strengths And Weaknesses:**

1, Computational Cost: Generating and evaluating groups for large datasets (e.g., ImageNet) can be computationally expensive.
2, Out-of-Distribution Data: The binarization of groups may create out-of-distribution data, although the paper argues that modern Transformers are robust to this.
3. Limited Human Evaluation: The human distinction task is conducted on a small subset of images (10 examples), which may not be representative.

**Questions For Authors:**

See above comments.

**Relation To Broader Scientific Literature:**

I believe it is crucial to investigate self-attributing in terms of groups of features rather than individual tokens or features. However, AutoGnothi [1] already achieves faithful interpretability without compromising performance. It is therefore important for the authors to provide a deeper discussion of related work and offer a more thorough comparison of the results.

**Theoretical Claims:**

1. Theoretical Claim 1: Per-feature SANNs have a lower bound on error that grows with the number of features, especially for correlated data.

Support: Theorems 2.3 and A.2 prove that per-feature SANNs cannot model simple polynomial functions with correlated features. The error grows exponentially with the number of features (Figure 2).

2. Theoretical Claim 2: Group-based SANNs can achieve zero error if the groups align with the underlying correlations in the data.

Support: Theorems 2.4 and A.5 show that group-based SANNs can perfectly model complex polynomials with zero insertion and deletion errors.

---

> ### Author Rebuttal · Authors · 2025-04-01
>
> Thank you for your feedback. We provide an additional PDF at https://github.com/icml2025-3311/icml2025-3311/blob/main/ICML2025_3311_rebuttal.pdf for additional figures and will respond to your questions below.
>
> ## Comparison with Additional Shapley-Based Baselines
> We first would like to clarify a key distinction between SANNs and the referenced line of Shapley-based methods.  Specifically, **these Shapley-based methods (SBMs) are not Self-Attributing Neural Networks (SANNs)**. Instead, these methods are faster approximations of the SHAP baseline that we already compare to. The goal of these SBMs is to efficiently approximate post-hoc explanation Shapley values, with later works (FastSHAP, ViT-Shapley, Auto-Gnothi) training additional surrogate models to estimate Shapley values. AutoGnothi, for example, trains multiple side networks to generate approximations of Shapley explanations. **These SBM explanations are crucially never passed into the backbone to make a prediction.** In contrast, in a SANN, the features in an explanation are directly passed into the backbone to guarantee that predictions rely solely on the explanation. Therefore, the explanations of AutoGnothi and other Shapley-based methods are more similar to post-hoc explanations and do not fall into the category of SANNs. This fact is reflected in SBM experiments: all the papers of FastShap, ViT-Shapley, and Auto-Gnothi only compare to post-hoc explanation methods and do not compare to any SANNs, as a direct comparison is not possible.
>
> In our paper, we provide comparisons to SANN variants of post-hoc methods. Since we already compared to a SANN variant of SHAP (in Table 1, denoted SHAP-F), for completeness, we ran analogous comparisons with SBMs using their released codebases. Since the suggested methods involve training additional side networks, for fair comparison, we ensured that all baselines had equal or greater training resources than SOP. **We show the results in Table 1 in the additional PDF,  where we find that their performance as a SANN is actually even worse than SHAP, and consequently worse than SOP.** Intuitively, this result is not too surprising since these methods are faster but more inaccurate approximations of SHAP, trading off accuracy of the Shapley value for speed. KernelSHAP was too expensive to be fairly run as a baseline, as it requires a minimum number of forward passes at least equal to the number of features to find the solution to their loss function in Theorem 2 in their paper and code [2], making it an order of magnitude more expensive than all other baselines.
>
> ## On Computational Cost of SOP
> Since the reviewer brings up computational cost as a potential weakness, we point out that **our SANN experiments are orders of magnitude larger scale than those of the referenced SBMs**. The AutoGnothi and ViT-Shapley papers conduct experiments on small scale datasets such as ImageNette, which only has 10 easily distinguishable classes where it is easy to get 100% accuracy. In contrast, **we train SANNs on full ImageNet (1000 classes!), where SOP achieves SOTA performance with only one epoch of training.** Since we are evaluating SBMs in a significantly larger setting than originally proposed,  we checked the accuracy of surrogate models to ensure these baselines were trained properly. We confirmed that the resulting surrogate models trained on masked inputs have comparable accuracy on ImageNet to other work in the literature [1, Figure 10] that also trains models on similarly masked inputs, so we are certain these baselines were sufficiently trained.
>
> If the reviewer is still concerned about cost, we have also included an ablation study (Figure 1 in the additional PDF) showing that SOP performance on ImageNet saturates at just 5 groups, meaning we can further reduce inference costs by an additional factor of 4 beyond what we originally reported.
>
> [1] Jain et al. Missingness Bias in Model Debugging. ICLR 2022.
>
> ## Human Distinction Task With More Samples
> First, we point out that **our human distinction task follows the exact same protocol as the original HIVE paper published at ECCV** [2a], which uses 10 examples for evaluating human distinction tasks [2b]. We believe this to be partly because evaluating the task on a single example requires significant human effort, with the original study consisting of hundreds of evaluator tasks. **We have increased the study to 50 examples,** which amounts to thousands of evaluator tasks. In summary, we found results consistent with what we originally reported previously (that SOP is among the best, but that the distinction test cannot distinguish between the best), but with slightly smaller error bars and smaller differences between methods. The expanded study is shown in Figure 3 in the additional PDF.
>
> [2a] Kim et al. HIVE: Evaluating the Human Interpretability of Visual Explanations. ECCV 2022.
>
> [2b] https://github.com/princetonvisualai/HIVE/blob/main/materials/HIVE_suppmat.pdf

---

> > ### Comment · Reviewer_rZvr · 2025-04-07
> >
> > Thanks for your rebuttal. I would like to point out that AutoGnothi is actually an SANN, which simualtanesouly outputs the prediction and its explanation. I suggest the authors compare the results with AutoGnothi.

---

> > > ### Author Response · Authors · 2025-04-08
> > >
> > > ## Reviewer’s mischaracterization of AutoGnothi vs. SANNs
> > > **A network that simultaneously produces a prediction and explanation is not necessarily a SANN.** SANNs explicitly use feature subsets to make predictions, formalized as $f(x) = \sum_{i=1}^m \theta(x)_i h({x_G}_i)$ (Definition 2.1). This SANN framework aligns with prior work [1,2,3]. AutoGnothi generates predictions and explanations separately by using different heads on the last hidden states [4]. However – **this explanation is never used for feature selection to make the prediction, as stated in the AutoGnothi paper**. While AutoGnothi simultaneously produces explanations and predictions, it does not fall under the SANN framework and is therefore not a SANN.
> > > This was explained in our [initial rebuttal](https://openreview.net/forum?id=r6y9TEdLMh&noteId=uKLJz7H3DW).
> > >
> > > If the reviewer can concretely explain the original AutoGnothi model can be formalized as a SANN then we would be happy to categorize it as such.
> > >
> > > ## Initial rebuttal contains requested comparisons
> > > **Requested comparisons to AutoGnothi, FastShap, and ViT-Shapley have already been reported on their SANN variants.** As we previously explained in detail in the [initial rebuttal](https://openreview.net/forum?id=r6y9TEdLMh&noteId=uKLJz7H3DW), these methods are faster, less-accurate approximations of SHAP. **SOP outperforms SHAP and all requested baselines**. Nonetheless, we are happy to include these comparisons in the final version.
> > >
> > > [1] David Alvarez-Melis, Tommi S. Jaakkola. Towards Robust Interpretability with Self-Explaining Neural Networks. NeurIPS 2018.
> > >
> > > [2] Brendel, W. and Bethge, M. Approximating CNNs with Bag-of-Local-Features Models Works Surprisingly Well on ImageNet. ICLR 2019.
> > >
> > > [3] Jain, S., Wiegreffe, S., Pinter, Y., and Wallace, B. C. Learning to Faithfully Rationalize by Construction. ACL 2020.
> > >
> > > [4] Shaobo Wang, Hongxuan Tang, Mingyang Wang, Hongrui Zhang, Xuyang Liu, Weiya Li, Xuming Hu, Linfeng Zhang. Gnothi Seauton: Empowering Faithful Self-Interpretability in Black-Box Transformers. ICLR 2025.

---

### Official Review · Reviewer_GA2V · 2025-03-18

**Overall Recommendation:** 4

**Summary:**

The paper addresses the limitations of existing self-attributing neural networks (SANNs) in high-dimensional data. The authors theoretically prove a lower bound on the error of per-feature SANNs and demonstrate that group-based SANNs can overcome this limitation. The main algorithmic contribution is the Sum-of-Parts (SOP) framework, which transforms any differentiable model into a group-based SANN. SOP achieves state-of-the-art performance for SANNs on vision (ImageNet-S, CosmoGrid) and language (MultiRC) tasks. The learned feature groups are shown to be interpretable through quantitative metrics (performance at different sparsity levels, faithfulness) and semantic metrics (semantic coherence, human distinction). Furthermore, the paper demonstrates the utility of SOP explanations in model debugging and cosmological scientific discovery.

**Claims And Evidence:**

Overall, the claims are well-supported by evidence; however, some minor points may require further clarification or additional evidence.

1. Performance : While SOP achieves strong results on insertion metrics, its performance on deletion metrics is somewhat weaker compared to certain baselines like Archipelago or FRESH in specific cases. Although the authors explain this behavior as natural due to SOP's reliance on multiple feature groups rather than single groups, this aspect could benefit from further empirical exploration to fully justify this claim.

2. Overhead : The authors propose SOP as a flexible and model-agnostic framework; however, they do not extensively discuss computational overhead or scalability implications clearly. Given that SOP involves dynamic attention-based group generation and multiple forward passes through pre-trained models per input example, computational efficiency could be a concern in practical applications.

**Essential References Not Discussed:**

The paper does a thorough job of discussing related works in the context of self-attributing neural networks and interpretability.

**Experimental Designs Or Analyses:**

The experimental designs and analyses are well-executed and provide convincing evidence for most claims made in the paper. However there are some minor issue where where further validation could strengthen the work.

1. Pre-Trained Backbones: SOP uses pre-trained models (e.g., ViT, BERT), while some baselines (e.g., BagNet) do not. Hence it is ideal to compare SOP to baselines using the same pre-trained backbone for fairness.

2. Computational Efficiency : SOP’s dynamic group generation and sparse attention may increase computational cost compared to baselines. Hence the authors should include a detailed analysis of training/inference times and memory usage.

3. Generalizability: The proofs focus on polynomial functions, which may not capture all real-world data patterns (e.g., non-polynomial interactions). I would like the authors to comment here.

**Methods And Evaluation Criteria:**

The proposed methods and evaluation criteria make sense for the problem at hand. However I have some doubts and would like the authors comment on them.

1. Group Diversity: Do you think multi-head attention may produce redundant groups? If yes, methods like diversity regularization could help?

2. Deletion Bias: Deletion tests generally favor models using fewer features, slightly disadvantaging SOP’s multi-group approach.

**Other Comments Or Suggestions:**

Typos:

1. L89: "anddifferent" -> "and different"
2.L211: "in and Figure" -> "in Figure"

**Other Strengths And Weaknesses:**

The paper makes a compelling contribution to interpretable ML by addressing the trade-off between performance and faithfulness in SANNs. But some minor weaknesses exist according to me.

1. Computational Cost: While the paper demonstrates strong performance, it does not extensively discuss the computational cost associated with the SOP framework. The use of attention mechanisms, especially multi-headed self-attention, can be computationally intensive. Providing an analysis of the training and inference time complexity and potentially comparing it to other SANN methods would be valuable for understanding the practical feasibility of SOP.

2. Hyperparameter Sensitivity: The paper mentions using a sparsity level of τ=20% for the group generator but does not delve deeply into the sensitivity of the results to different hyperparameter settings, such as the number of groups (m) or the sparsity level. An analysis of how these hyperparameters affect the performance and interpretability of SOP would strengthen the paper.

3. Interpretability of Learned Groups: While the paper presents quantitative and qualitative evidence for the interpretability of the learned groups, further visualization and analysis of the actual feature groups discovered by SOP could provide more intuitive insights. Understanding what kind of feature groupings the model learns for different tasks and classes could further enhance the understanding and trust in the explanations provided by SOP.

4. Cosmological Evaluation Metrics: The threshold-based purity metrics for CosmoGrid rely heavily on domain expertise. While justified by collaboration with cosmologists, these metrics may be less accessible or transparent to non-expert readers without additional context or sensitivity analyses.

**Questions For Authors:**

It would be great if the authors consider below recommendations to improve the paper:
1. Provide a computational complexity analysis and discuss optimizations for scalability.
2. Include a discussion of potential biases inherited from pre-trained backbones and mitigation strategies.

Overall, the paper presents significant original contributions through rigorous theory, innovative methodology, strong empirical validation across multiple domains, and practical utility in scientific discovery. Minor weaknesses include limited exploration of computational overhead, and brief explanations regarding certain evaluation metrics, which I mentioned before. These weaknesses do not substantially undermine the paper's core contributions but indicate areas where additional clarity or evidence could further strengthen the work.

**Relation To Broader Scientific Literature:**

SOP's key contributions lie in its theoretical insights into the limitations of per-feature SANNs, its novel framework for end-to-end learning of semantically coherent feature groups without supervision, its state-of-the-art performance among SANNs, and its rigorous validation of interpretability and utility in diverse applications. These advancements significantly contribute to the ongoing research efforts in the field of interpretable machine learning.

**Theoretical Claims:**

The theoretical claims are well-supported by rigorous proofs and empirical validation, particularly for Theorems 2.3 and 2.4. The authors also posit conjectures (e.g., Conjecture A.1 and A.2) suggesting exponential growth of insertion/deletion errors for monomials and binomials as feature dimensions increase. These conjectures are supported by empirical fits to numerical results but lack formal proofs.

---

> ### Author Rebuttal · Authors · 2025-04-01
>
> Thank you so much for the valuable and encouraging review! We provide an additional PDF at https://github.com/icml2025-3311/icml2025-3311/blob/main/ICML2025_3311_rebuttal.pdf for additional figures.
>
> ## Explaining the Deletion Performance of SOP.
> To explain how SOPs usage of multiple groups can affect the deletion, we show the average deletion curves for SOP in Figure 2 in the linked PDF. We see that even as we delete features from groups, SOP is able to maintain relatively high performance in some cases, resulting in a worse deletion score. **Such behavior is natural because the training objective in SOP encourages the group selector to select highly predictive groups, and multiple groups can compensate for the information missing in another group.**
>
> ## What is the computational complexity of SOP?
> The cost of SOP (running time and memory) is dominated by the forward passes through the backbone predictor $h$, which can be seen in the definition of $f$ in Section 3. **Therefore, when using $m$ groups, the cost of an SOP forward pass is equivalent to $m$ forward passes through the backbone.** The costs of the group generator and the group selector are negligible in comparison, since each one is a much smaller attention module on sequences of $m$ inputs. We will make this cost explicit at the end of Section 3.
>
> In our experiments, we control computation fairly across all baselines, which is discussed at length in Appendix C.2 & C.3. All methods can use at most 20 forward passes per inference with the exception of Archipelago which requires quadratic forward passes (Appendix C.3 line 1531-1533). For SOP, this amounts to **m=20 groups and thus 20 forward passes** per inference (Appendix C.2.2 line 1462-1463). SOP can match or beat all other baselines with equal compute.
>
> A detailed summary of computation is in Table 2 in the linked PDF, which we will add to Table 2 of the main paper to make this information more visible.
>
> Lastly, the training needed to fine-tune SOP modules (Appendix C.2.2 line 1464-1465) are minimal compared to training the backbone (e.g. one epoch for ImageNet).
>
> ## Questions about Group Diversity.
> Multihead attention and explicit diversity regularization could potentially change group diversity. We found that **simply using more attention heads in the group generator actually helped create more diverse groups,** an observation that we briefly mentioned in Appendix C.2.2 line 1462-1462. Concretely, we observed that using four heads in the group generator results in 32% fewer overlapping patches in groups than one head on ImageNet.
>
> ## Comparing SOP to Baselines with the Same Backbones.
> We confirm that SOP is already compared to baselines with the same backbones whenever possible. Specifically, **we compared 10 model-agnostic baselines that use the exact same backbone model as SOP**, and 4 baselines that are architecture specific. This is summarized in the main paper in Table 1 (see “Model-Agnostic” column) and discussed at length in Appendix C.3. We included architecture-specific baselines as readers wanted to see such comparisons.
>
> ## Generalizability of Polynomials in the Theory.
> We note that our theory tackles polynomials in the most general setting, with no restrictions on the type or degree of polynomials, and therefore has broad implications. Thanks to the Stone-Weierstrass theorem, every continuous function can be uniformly approximated by a polynomial function. **Since our result applies to any polynomial, it also applies to those that uniformly approximate real-world data patterns.**
>
> ## Sensitivity to Sparsity / Number of Groups.
> - *Sparsity Analysis*. **We have already done this sparsity analysis in Section 4.2 RQ2 (Figure 3)**, with further analysis in Appendix C.4.2 (Figure 18).
>
> - *Number of Groups Analysis*. In Figure 1 in the linked PDF, we measure how the number of groups affects accuracy for ImageNet. Naturally, **having more groups improves accuracy**, and it saturates at 5 groups, suggesting that SOP can actually be 4x more efficient than originally reported.
>
> ## Further Visualization & Analysis of Groups.
> **We have already included visualizations** in Figures 5+7 in the main paper as well as Figure 11~15 in the appendix for more visual examples, and **RQ5 and RQ6 do a semantic analysis of said groups**. Furthermore, **RQ7 and RQ8 provide insights on how different features groups learned with SOP affect predictions**.
>
> ## More Context for Cosmology Study.
> We referenced additional detailed and accessible cosmology background in Appendix D, where we introduce the problem and discuss the kinds of findings and groups cosmologists find interesting.
>
> ## Biases in Pretrained Models.
> Pretrained models are known to encode a range of social biases including racism and sexism. One could utilize algorithms that edit models for fairness or debiasing models. Such biases equally affect all baselines in our work using the same pretrained model.

---

> > ### Comment · Reviewer_GA2V · 2025-04-08
> >
> > Thank you for your detailed response, which has effectively clarified my doubts. I appreciate the effort you put into addressing my concerns.
> >
> > Based on your reply, I have adjusted my rating accordingly.

---

> > > ### Author Response · Authors · 2025-04-09
> > >
> > > Thank you for your encouraging comment and for updating your recommendation. We appreciate your feedback on improving our paper and will incorporate all important points from our discussion into the revision.

---

### Decision · Program_Chairs · 2025-05-01

**Decision:**

Accept (poster)

**Comment:**

The authors propose Sum-of-Parts (SOP)  framework for SANN. The reviewers find the work solid in both theoretical and empirical results.  The concerns from the reviewers are properly addressed.